# Even moderate geomagnetic pulsations can cause fluctuations of foF2 frequency of the auroral ionosphere

Nadezda Yagova[1], Alexander Kozlovsky[2], Evgeny Fedorov[1], and Olga Kozyreva[1]

[1]Schmidt Institute of Physics of the Earth, Moscow,Russia
[2]Sodankylä Geophysical Observatory, Sodankylä, Finland

**Correspondence:** N. Yagova(nyagova@ifz.ru)

**Abstract.** The ionosonde at the Sodankyla Geophysical Observatory (SOD, 67 N, 27 E, Finland) routinely performs vertical sounding once per minute, such that makes possible studying fast ionospheric variations at a frequency of the long-period geomagnetic pulsations Pc5-6/Pi3 ($1-5$ mHz). Using the ionosonde data from April 2014 - December 2015 and collocated geomagnetic measurements, we have investigated a correspondence between the magnetic field pulsations and variations of the critical frequency of radio waves reflected from the ionospheric F2 layer (foF2). For this study, we have developed a technique for automated retrieval the critical frequency of the F2 layer from ionograms. As a rule, the Pc5 frequency band fluctuations in foF2 were observed in the daytime during quiet or moderately disturbed space weather conditions. In many cases (about $80\%$), the coherence between the foF2 variations and geomagnetic pulsations was low. However in some cases (specified as "coherent") the coherence was as large as $\gamma^2 \geq 0.5$. Favorable conditions for the coherent cases are enhanced auroral activity (6-hour maximal AE $\geq 800$ nT), high solar wind speed ($V > 600$ km/s), fluctuating solar wind pressure, and northward interplanetary magnetic field. In the cases when the coherence was higher at shorter periods of oscillations, the magnetic pulsations demonstrated features typical for the Alfven field-line resonance.

## 1 Introduction

Ionospheric variations at the frequency corresponding to simultaneous geomagnetic pulsations were observed in the total electron content (TEC) and have been interpreted as a modulation of the ionosphere by magnetospheric processes (e.g. Davies and Hartmann (1976); Okuzawa and Davies (1981)). The effect of the ionosheric modulation at a frequency of Pc3-4 geomagnetic pulsations ($6.7-80$ mHz) was found at mid and low latitudes (Davies and Hartmann, 1976; Okuzawa and Davies, 1981). Davies and Hartmann (1976) reported on pulsations in TEC, recorded during non-disturbed conditions and associated with Pc3-4s on the ground. Later, Okuzawa and Davies (1981) confirmed the correspondence between ground and TEC variations in the Pc3-4 frequency range, but they obtained the maximal probability under disturbed conditions.

Particle precipitation at auroral latitudes is one of the most intensive processes of the magnetosphere-ionosphere coupling. It is modulated by intensive ultra low frequency (ULF) waves and this was a topic of numerous studies. Clear features of Pc5 pulsations in the Doppler velocities of the F-layer ionospheric irregularities were found by Ruohoniemi et al. (1991) using coherent-scatter radar observations. Wright et al. (1997) analyzed ionospheric signatures of auroral Pc4-5s with Doppler sounding. They have shown that large-scale pulsations correlated in the ionosphere and on the ground demonstrated azimuthal wave numbers $m$ typical for ground Pc4-5s and field line resonance (FLR) features. Particle populations responsible for generation of small-scale (high $m$) waves and their ionospheric signatures were studied by Baddeley et al. (2005). The substorm Pc5 waves with intermediate $m$ registered simultaneously in ion flux in the magnetosphere and in radar observations were reported by Mager et al. (2019). James et al. (2016) showed that the energy of precipitating particles and ULF wave number were controlled by the distance from the substorm epicenter.

Publications devoted to Pc5 signatures in total electron content (TEC) are not so numerous, as radar ones. Watson et al. (2015) presented the large-amplitude TEC variations in the Pc5-6 frequency band detected by GPS receivers. They suggested that these TEC variations were associated with the compressional mode magnetospheric waves. The corresponding pulsations were also manifested in the magnetic field on the ground with two spectral maxima at about 0.9 mHz and 3.3 mHz. This event was observed in the afternoon after a steep increase of solar wind (SW) dynamic pressure up to almost 20 nPa. It was also followed by a modulation of electron flux measured at geostationary orbit. An important feature of the observed oscillations is that the amplitude of TEC variations is higher than that found by Pilipenko et al. (2014a), although the geomagnetic pulsations on the ground were not so intensive.

Vorontsova et al. (2016) observed the effect of TEC modulation by ULF waves at low latitudes far away from the resonant L-shells and the zones where kinetic modes can occur due to wave-particle interaction. This has allowed to identify the observed ULF waves as fast magnetosonic mode. Kozyreva et al. (2019) investigated the Pc5 frequency band oscillations in the ionosonde data at an auroral station and found an intriguing effect of the ionopheric oscillations at a second harmonic of the simultaneously observed geomagnetic pulsations.

Pilipenko et al. (2014a) have found the effect of ionosphere heating in the F-layer by an intense magneto-hydrodynamic (MHD) wave at the recovery phase of a magnetic storm. It was not associated with noticeable electron precipitation. The TEC variations in this event was then studied by Pilipenko et al. (2014b) and Pc5 related TEC variations were retrieved. Pilipenko et al. (2014b) suggested several mechanisms of the TEC modulation by ULF waves associated with Pc5 pulsations and estimated the related amplitudes of TEC variations. For some of them no detectable variations of TEC were found under realistic Pc5 amplitudes. Meanwhile, such mechanisms as Joule heating provide detectable amplitudes for intensive Pc5s.

The efficiency of the TEC modulation by ULF waves depends on the modulation of the particle flux in the magnetosphere. However, long-term measurements at geostationary orbit are available for electrons at energies $E > 30$ keV. A data deficit at lower energies can be partly compensated by numerical modelling. Buchert et al. (1999) modeled variations of ionospheric Hall and Pedersen conductivities associated with ULF modulation of the electron flux at energies about several keVs. These results correspond to the E-layer of the ionosphere, i.e. the electron energy is still too high to control foF2 fluctuations. Recently, Van

Allen probes measurements (Ren et al., 2019) confirmed the influence of ULF waves in Pc4 and Pc5 frequency ranges on the electrons at energies about $10^2$ eV with maximal occurrence at $L$ from 5.5 to 7.

Watson et al. (2015) showed experimentally that high amplitudes of TEC variations can be observed under lower Pc5 amplitudes than follows from the analysis of ULF wave modulation of the ionosphere without a particle flux modulation.

Thus, a question arises about a statistical relationship between geomagnetic and ionospheric pulsations. Till now, a role of MHD waves in a wide range of amplitudes in variations of foF2 critical frequency has not been studied sufficiently.

In the present paper we attempt to study statistics of the variations of foF2 critical frequency at the Pc5/Pi3 frequencies and simultaneous geomagnetic pulsations in the same frequency range.

## 2 Data of observations and their processing

### 2.1 Data

Data of the ionosphere sounding were obtained from the Sodankylä Geophysical Observatory (SOD, 67.3° N, 26.7° E). The SOD ionosonde routinely performs vertical sounding once per minute. A detailed description of the ionosonde can be found in (Kozlovsky et al., 2013). The SOD magnetometer is included in the IMAGE magnetometer network (Taskanen, 2009), and data of the three components of the geomagnetic field are available with 10 s sampling rate. For the analysis of the spatial distribution of geomagnetic pulsations we also use data from another IMAGE magnetometer in Masi (MAS), which is located nearly at the same geomagnetic longitude, but at higher latitude (Table 1).

The space weather data were obtained from the OMNI database at http://cdaweb.gsfc.nasa.gov. We used data of the interplanetary magnetic field, speed and dynamic pressure of the solar wind, re-calculated to the sub-solar point of the magnetosphere (Bargatze et al., 2005), and the indices of geomagnetic storm and auroral activity, namely the disturbance storm time (Dst) and Auroral Electrojet (AE) indices.

We have looked through the ionosonde and SOD magnetometer data from April 2014 through December 2015 and visually selected intervals for the further analysis according to the following criteria:

– The F trace is clearly seen in the ionograms, such that the foF2 critical frequency can be retrieved;

– Geomagnetic pulsations Pc5 or Pi3 are seen in the northward ($B_X$) component and their peak-to-peak amplitude exceeds 5 nT during at least 2 hours in the daytime, between 8 and 14 UT corresponding to 11 - 17 magnetic local time, MLT, at SOD

A list of selected intervals is given in Table 2.

### 2.2 foF2 automatic detection from ionograms

Although visual scaling of the foF2 values from ionograms with clearly identified frequency traces (F traces) is easy, the studies of high-frequency variations require scaling of many ionograms, such that one needs an automated procedure for that.

The difficulties of this procedure are caused by variability of the intensity of reflected signals, background noise, sporadic layers and irregularities, broadcast interference, etc. Because of these reasons, techniques of the automated foF2 detection can be unstable, even in the cases when visual detection is possible.

Note, that in the problem of ionospheric density fluctuations two different frequency ranges are involved, namely frequencies of the ionosphere sounding which is about several MegaHertz and frequencies of geomagnetic and foF2 pulsations in the milliHerz range. We use different notations, $F$ and $f$, for them.

In the present study, we used a method based on the approximation of the F trace in a wide range of altitudes to reduce the influence of gaps and intensity peaks of reflections at some frequencies.

The F trace (i.e., the curve showing dependence of the frequency of reflected wave on the virtual height of reflection) is characterized by a near linear growth at low height with gradual transition to saturation at the critical frequency (Figure 1). The reflection intensity in Figure 1 is given in dB, whereas in the calculation a linear scale is used (voltage in arbitrary units).

We approximate this dependence by a Lorentzian function such as

$$F(h) = F_1 + \Delta F \frac{k(h - h_1)^\alpha}{k(h - h_1)^\alpha + 1} \tag{1}$$

The approximation was made above a starting height $h_1 = 235$ km. At this altitude $F_1 = F(h_1)$ and $F_2 = F_1 + \Delta F$ is the frequency limit at $h \to \infty$. Actually, foF2 is close to $F_2$, but a minor difference between these two values indicates that a non-zero positive derivative in the $F(h)$ dependence exists at all the altitudes. Parameters $F_1$, $\Delta F$, $k$, and $\alpha$ were obtained in the course of a fitting procedure described below. The trace was determined as a curve where the following two conditions were fulfilled:

- Intensity of the reflection, $I$, at the trace is high $I \geq I_b$, where $I_b = 4 \cdot 10^4$

- Ratio of the signal intensity at the trace to that above it, $R$, is high ($> 3$)

For the four-factors fitting, a 9-point iteration procedure was applied to maximize a parameter $K_t^2 = cI^2 + (1 - c)R^2$ in the space of parameters $K_t(x_0, x_0 - \Delta x_i, x_0 + \Delta x_i)$, where $c$ is a constant between 0 and 1, $x$ is a point in the space of parameters, and $i$ identifies the parameter. An initial approximation was taken from the database created manually for several typical types of the $F(h)$ dependence. After that, the foF2 was determined as $F$ given by eq.(1) at the heights at which it weakly depends on $h$. The other requirement was a continuity of the time dependence foF2($t$). The threshold value for the time derivative of foF2 was estimated from the foF2 standard deviation, obtained from $N$ previous instants. In the present version of procedure $N = 10$ and the maximal $t - t_0$ difference equal to 2 standard deviations was used. For $t > t_1$, the set of parameters calculated at the previous step was taken as an initial approximation. If the iteration gave a value of foF2, for which the difference from the previous values exceeded the threshold value, another initial approximation was taken from the database, and the procedure was repeated. If all the initial approximations gave values standing far from the previous ones, this data point was excluded, and the iteration procedure started from the next time instant. Examples of the $F$ trace approximation are shown by white curves in Figure 1 for three ionograms obtained on 24 October 2014. (Note that the ionograms are rotated by 90° with respect to

traditional presentation.) For the example shown in Figure 1a the critical frequency foF2 obtained from eq.(1) and parameters $F_1$, $F_2$ and $h_1$ are indicated. For this case $\alpha = 1.37$ and $k = 6.5 \cdot 10^{-3}$ were obtained.

The continuity condition allows to reduce effects of multiple reflections and bifurcations. ASCII files of the data such as retrieved foF2 and parameters of the approximation, and plots of foF2 time series are available as supplementary materials. In all the cases, the results of the automated procedure were tested visually for each tenth data point. The selected intervals form the database for the analysis. As an example, Figure 2 shows a diurnal variation of the critical frequency retrieved by the automated procedure for the case on 3 January 2015.

**2.3   Pre-processing, statistical and spectral analysis**

We have analyzed pulsations in the meridional component of the geomagnetic field in association with variations of foF2 critical frequency. In the present study we consider geomagnetic pulsation in the $1-5$ mHz frequency band with no separation to classes Pc5, Pc6, and Pi3. Such a separation is often vague, and an automated identification is hardly possible in practice. Moreover, morphological types of pulsations, especially with large azimuthal wave numbers are not identical, as measured in

the magnetosphere and on the ground (e.g. Vaivads et al. (2001)). Besides, an analysis of the numerous Pc5-6/Pi3 subclasses according to Saito (1978) would be too cumbrous. Thus, we study inter-relations between all geomagnetic and foF2 variations in the frequency range of the Pc5-6/Pi3 pulsations and investigate their dependencies on the space weather parameters.

We have studied how the occurrence of foF2 fluctuations in Pc5-6/Pi3 range and their coherence with geomagnetic pulsations depend on the magnetic local time (MLT), parameters of geomagnetic pulsations, and space weather conditions.

The Dst and AE indices were used as indicators of geomagnetic storms and auroral substorms, respectively.

The interplanetary parameters controlling the Pc5-6/Pi3 activity are the vertical component of the interplanetary magnetic field (IMF $B_Z$), solar wind velocity $V$, and fluctuations of the solar wind pressure $P_{SW}$ (Baker et al., 2003).

For further analysis, al the intervals are classified into 5 groups.

– All intervals in April 2014 - December 2015. For them, space weather parameters and geomagnetic activity indices are
analyzed (all intervals, group 1).

– The intervals, for which spectra of geomagnetic variations are calculated (Pc5-6/Pi3 intervals, group 2).

– The intervals, for which foF2 spectra can be calculated (foF2 intervals, group 3).

– The intervals in which the coherence of foF2 variations with geomagnetic pulsations exceeded certain threshold (coherent foF2-$b_X$ intervals, group 4).

– Coherent pulsations with a coherence maximum at the high-frequency flank of the band ($f_\gamma > 2.7$ mHz, group 5).

The space weather parameters, activity indices, and geomagnetic data are available for most of the intervals during April 2014 -December 2015. To avoid possible influence of seasonal and diurnal variations, we used data from the same time, 8-14 UT, when foF2 data can be available around the year. We used this classification to compare distributions of the space weather

parameters for all the intervals (group 1) with those for the foF2 intervals (group 3) and coherent foF2-$b_X$ intervals (group 4). Also, we have investigated which conditions are favorable for observations of the variations in foF2 and for the coherent foF2-$b_X$ events.

One-minute time resolution of the data of foF2 allows the cross-spectral analysis with the Pc5-6/Pi3 geomagnetic pulsations. For that, these datasets power spectral density (PSD) and cross-spectral parameters were estimated using the Blackman-Tukey method (Kay, 1988). The cross-spectra have been calculated between the foF2 variations on one hand, and geomagnetic pulsations on the other hand. For the intervals with high spectral coherence, $\gamma^2$, a phase difference $\Delta\varphi$ was estimated.

Spectra were calculated for nearly 64-min ($N_p = 64$ points) intervals with a 5-min shift. Such a time step allows to detect short living Pi-type pulsations and an equal length of the time intervals allows to classify the intervals according to the parameters of their spectra (such as maxima of PSD or coherence, or frequency of the PSD maximum) and use them for statistical analysis.

A total number of the intervals in group 3 is 2764. For group 4, it is 448. Note, that the pulsations are not independent because some intervals are overlapping, and the numbers of non-overlapping intervals are 240 and 114 for groups 3 and 4, respectively.

Parameters of the spectral analysis are selected as a compromise between the frequency resolution and the dispersion of the spectral estimates. We used a 16 point ($M = N_p/4$) window. A dispersion of the smoothed coherence spectra can be calculated as

$$var(\gamma^2) = \frac{c}{K}(\gamma^2)(1-\gamma^2)^2, \tag{2}$$

where $K = N_p/M$, and $c$ is a constant depending on the spectral window. Eq. (2) shows that the dispersion of coherence depends on its absolute value (Jenkins and Watts, 1969). It goes to zero for high coherence and it is about the dispersion of non-smoothed spectra for low coherence values. We used $\gamma_b^2 = 0.5$ as a threshold value of coherence in the present study, which corresponds to a dispersion of 0.074. This means that for $\gamma_b^2 = 0.5$, $\gamma^2$ exceed 0.25 at 70% confidence level.

## 3 Results

### 3.1 foF2 variations and geomagnetic pulsations at SOD

#### 3.1.1 Examples

Below we present two examples of the foF2 and geomagnetic variations simultaneously recorded at SOD. They are characterized by a high foF2-$b_X$ coherence, however the amplitudes of geomagnetic pulsations were essentially different in these two cases, 4 nT and 40 nT, respectively. Geomagnetic and foF2 pulsations recorded on 11 March (Day 70) 2015 (event 1) are presented in Figure 3. The time series shown in Figure 3(a) are high pass filtered at 0.8 mHz. To discriminate the deviations of magnetic filed and foF2 from their non-disturbed values, the former are denoted as $\Delta$foF2 and $b$ for foF2 and magnetic field, respectively.

Peak-to-peak amplitudes of the geomagnetic pulsations and foF2 variations are about 8 nT and 0.05 MHz, respectively. Their maximal values are 12 nT and 0.1 MHz. The normalized PSD spectra, $PSD^*$, for both geomagnetic and foF2 pulsations, spectral coherence $\gamma^2$ and phase difference $\Delta\varphi$ are presented in Figure 4. $PSD^*$ of geomagnetic pulsations has two broad maxima at $f_1 = 2.3$ and $f_2 = 3.2$ mHz. The spectrum of foF2 variations has a maximum at a frequency $f = 3.8$ mHz. Spectral coherence ($\gamma^2 > 0.75$) in the low frequency part of spectrum $f < 2$ mHz, and a minor coherence peak with $\gamma^2 = 0.6$ near the $f_2$ frequency is seen. At $f < 1.6$ mHz where $\gamma^2 > 0.9$ the $\gamma^2$ dispersion does not exceed $6 \cdot 10^{-3}$, i.e. $\gamma^2 > 0.8$ at 83% confidence level. For $\gamma^2 = 0.75$ and $0.6$ the dispersion values are $0.027$ and $0.056$, respectively. Figure 5 shows the space weather conditions for event 1. Zero time in panels (a-e) corresponds to the start time of the interval (12:20 UT). It is seen that geomagnetic conditions were quiet, and Dst$> -20$ nT (Figure 5a) indicates that no geomagnetic storm occurred during at least four days before the event. However, the auroral activity was essential and maximal AE reached 500 nT (Figure 5b). This activation occurred after a negative (southward) $B_Z$ variation of about 20 nT (Figure 5d). For this event, SW speed $V$ was about 400 km/s (Figure 5c), and the SW dynamic pressure was about 4 nPa (Figure 5e). The $P_{SW}$ fluctuations are shown in more details in Figure 5f. Their peak-to-peak amplitude was about 0.7 nPa and their apparent period was about 5 minutes. This corresponds to a frequency $f = 3.3$ mHz, i.e. it approximately agrees with the $f_2$ frequency of pulsations at SOD.

The case on 11 July (Day 192) 2015 (event 2) is presented in Figures 6 and 7, which have the same format, as Figures 3 and 4 for the first event. Peak-to-peak amplitudes of geomagnetic and foF2 pulsations are about 80 nT and 0.08 MHz, respectively. A clear maximum at $f_1 \approx 2.5$ mHz is seen in both geomagnetic and foF2 PSD spectra (Figure 7a). At the second frequency $f_2 \approx 3.5$ mHz a maximum is seen only in foF2 variations, while in the geomagnetic pulsations this frequency is marked only as a plateau in the PSD spectrum. However, the both spectral maxima are seen clearly in the coherence spectrum (Figure 7b), and the phase difference is different for these two frequencies (Figure 7c).

Space weather conditions for this event are summarized in Figure 8, which has the same format as Figure 5. There were no a geomagnetic storm during 4 days before this event, which is indicated by the Dst exceeding $= -30$ nT (Figure 8a). Meanwhile, the auroral activity was high, namely, two auroral activations occurred at $\tau = -8$ and $-4$ hours hours with maximal AE$= 1300$ nT and 700 nT, respectively (Figure 8b). The first activation occurred after a 2-hours interval of negative IMF $B_Z$, while the second one was associated with a $B_Z$ turn from $-10$ to almost $+15$ nT (Figure 8d). For this event, $V$ was about 600 km/s (Figure 8c), whereas a maximal $P_{SW}$ was about 9 nPa, then dropped to 5 nPa and slowly decreased to about 3 nPa (Figure 8e). The peak-to-peak amplitude of $P_{SW}$ fluctuations during the interval shown in Figure 8f was about 0.35 nPa. Two types of variations can be found, namely fluctuations with an apparent period about 4.5 minutes and a series of steps at a periodicity of about 7 minutes. This correspond to frequencies 3.7 mHz and 2.4 mHz, i.e. near the frequencies of foF2 variations, registered at SOD.

### 3.1.2 Statistics

Figure 9 shows the MLT-dependence of the foF2 intervals (group 3) and relative occurrence of coherent events (group 4). One can see from this figure that the foF2 variations were detected in the near-noon and afternoon MLT sectors with a maximal

probability between 13 and 16 MLT. The lower panel shows the distribution of a relative occurrence of coherent ones (group 4). It varies in the range $0.13 - 0.21$ with a maximum near noon.

Figure 10a shows histograms for the frequencies of local PSD maxima in $b_X$ and foF2 spectra (groups 2 and 3, respectively). The geomagnetic pulsations demonstrate a maximum at 3.2 mHz, which corresponds to the frequency of the Alfven resonance at the $L$-shell of SOD. The frequency distribution of foF2 fluctuations has two maxima in the frequency bands centered at 2 and 3.9 mHz. Thus, the Figure has shown that the most probable frequencies of spectral maxima are different for foF2 and geomagnetic pulsations at SOD.

However, case studies show pulsations with maxima at the same frequencies in both foF2 and $b_X$. To check, whether this effect is a random co-incidence or the pulsations are interrelated, we have compared frequency distributions of the foF2 and geomagnetic pulsations for random (not equal, in a general case) time intervals with those recorded simultaneously. We have calculated a square difference, $\Delta_{f2} = (f_{F2} - f_b)^2$, where $f_{F2}$ and $f_b$ are the frequencies of foF2 and $b_X$ PSD maxima, respectively. The parameter $\Delta_{f2}$ was calculated for $f_{F2}$ and $f_b$ taken from the spectra, calculated at randomly selected and

simultaneous time intervals. To reduce possible influence of diurnal variation, for each $f_{F2}$ only those values of $\Delta_{f2}$ were used, which were obtained from $b_X$ spectra calculated at a random day at the same MLT with an 1-hour accuracy. Then, its average value was calculated. The difference in average values of $\Delta_{f2}$ for simultaneous and random intervals was quantified as a parameter $\delta = \log(\Delta_{f2,0}/\Delta_{f2,R})$, where $\Delta_{f2,0}$ and $\Delta_{f2,R}$ are the mean values of $\Delta_{f2}$ for simultaneous and random intervals, respectively. Negative values of $\delta$ indicate that the frequencies agree better for simultaneous, than for random intervals. The

frequency dependence of $\delta$ is shown in Figure 10(b). It is seen, that $\delta$ is negative in all the frequency bands, besides one centered at 2.5 mHz. A minimum is seen at $f = 3.2$ mHz, i.e. near the Alfven resonant frequency at SOD. Mostly negative values of $\delta$ indicate that the frequencies of geomagnetic and foF2 pulsations are closer to each other for the simultaneous time intervals than for the random ones. Thus, a process responsible for synchronization of the geomagnetic and foF2 pulsations should exist.

The inter-relation between geomagnetic and foF2 pulsations is manifested also in the coherent foF2-$b_X$ pulsations. Then, a question is arising, how the probability to detect a coherent foF2-$b_X$ pulsation depends on the parameters of geomagnetic pulsation and the space weather. To answer this question, we have studied three groups of parameters, namely:

- PSD, polarization and spatial distribution of geomagnetic pulsations;

- indices of geomagnetic storms (Dst) and auroral (AE) activity;

- the interplanetary parameters controlling geomagnetic activity.

First, we compared the parameters of geomagnetic pulsations at SOD for the Pc5-6/Pi3 intervals (group 2), foF2 intervals (group 3) and coherent foF2-$b_X$ intervals (group 4).

The results are presented in Figure 11. The PSD of geomagnetic pulsations are shown at panel 11a for the Pc5-6/Pi3 intervals (group 2) and foF2 intervals (group 3). The results for coherent foF2-$b_X$ pulsations (group 4) are almost the same as those for

the group 3 (not shown here). The the PSD in group 3 is higher than that in the group 2 at all frequencies. We think this is due to the selection criteria for foF2 intervals.

Polarization of the pulsations have been analyzed with a PSD ratio $R_{XY} = \text{PSD}_X/\text{PSD}_Y$, and the result is shown in Figure 11b. To test the hypothesis about possible influence of the Alfven field line resonance (FLR) on the foF2-$b_X$ interrelation, the group of coherent pulsations with a high frequency coherence maximum (group 5) is included in the analysis. The difference between Pc5-6/Pi3 intervals (group 2) and all the foF2 intervals (groups 3-5) is seen in a growth of $R_{XY}$ for the latter at $f > 2$ mHz. The slope of $R_{XY}$ is maximal for the group 5.

Following Baransky et al. (1995), we use a meridional PSD ratio in the resonance $b_X$ component calculated with the SOD-MAS station pair $\text{PSD}_{\text{SOD}}/\text{PSD}_{\text{MAS}}$, In contrast to group 2, meridional PSD ratios for groups 3-5 have maxima at $f = 3.2$ mHz. The curves for groups 3 and 4 are very similar, and effect is maximal for the group 5.

Thus, geomagnetic pulsations, recorded during foF2 intervals are polarized mostly along the meridian at the high-frequency flank of the spectrum. They also demonstrate a maximum in meridional PSD ratio at $f = 3.2$ mHz, i.e. near the Alfven resonant frequency at SOD.

The distribution for the groups 2-4 pulsations over the maximal PSD of $b_X$, $\text{PSD}_{\text{max}}$, is given in Figure 12. The upper panel shows the distribution for groups 2 and 3. The most probable $\text{PSD}_{\text{max}}$ values are $10^3 - 3 \cdot 10^3$ nT$^2$/Hz for the group 2, whereas for the group 3 they are $3 \cdot 10^3 - 10^4$ nT$^2$/Hz, i.e., 3 times higher. The relative occurrence of the group 4 weakly depends on $b_X$ PSD. It has a maximum in the same PSD band, in which the probability maximum is found. This value of PSD is typical, such as pulsations in this PSD band were observed in approximately $20\%$ of the spectra.

Statistics of the activity indices is presented in Figure 13. Left panels (13a,c) show distributions over Dst and AE indices for all, foF2 and coherent foF2-$b_X$ intervals (groups 1, 3, and 4). Right panels (13b,d) demonstrate group 4 relative occurrence.

Occurrence of a magnetic storms was indicated if a minimal Dst value dropped below $-25$ nT during 4 days before the analyzed interval. Geomagnetic and foF2 pulsations in such cases started immediately after the Dst minimum, i.e., at the main phase of the magnetic storm, and were observed up to 4 days after that, at the recovery phase of the storm. Probability maximum for the groups 3 and 4 is found for the Dst range $(-50, -25)$ nT, which corresponds to a weak geomagnetic storm. For the both groups 3 and 4 occurrence at Dst $(-75, -50)$ nT exceeds that for background (group 1). At this Dst level, more than $70\%$ of the pulsations were observed during 12 hours and longer after the Dst minimum, with a maximal occurrence 2 days after it, which corresponds to the recovery phase of a moderate geomagnetic storm (time delay distributions for different storm intensities are available in Figure S1 in Supplementary materials). Relative occurrence of coherent foF2-$b_X$ intervals group (4) is somewhat higher for moderate and low storm activity, Dst$> -75$ nT, than for Dst$< -75$ nT (Figure 13b).

The most important difference in the AE distributions of groups 1 and 3 was found for the AE values between 800 and 1600 nT (Figure 13c). For this interval, occurrence for group 3 is 0.1 against 0.17 for group 1. This is compensated by enhanced probability of AE in the range $(200 - 800)$ nT for the group 3, comparing to the group 1. Group 4 occurrence at AE within $(800 - 1600)$ nT is about the same as for the group 1. This is also emphasized in the enhanced group 4 relative occurrence (0.26) for this level of AE against 0.17 at AE$< 400$ nT (Figure13d).

Summarizing, we can say that intervals after moderate magnetic storms are favorable for foF2 fluctuations in Pc5-6/Pi3 range. However, no essential difference in Dst was found between groups 3 and 4. Fluctuations of foF2 were registered pre-

dominantly under a moderate auroral activity. Meanwhile, higher levels of auroral activity, namely $(800 - 1600)$ nT AE, are favorable to register coherent foF2-$b_X$ pulsations.

The statistical results for the interplanetary parameters are presented in Figure 14. The IMF $B_Z$ controls the energy input from the solar wind to the magnetosphere. Generally, all auroral phenomena, including Pc5-6/Pi3 pulsations, are more intensive and occur at lower latitudes during negative IMF $B_Z$. A positive IMF $B_Z$ causes enhanced activity at higher latitudes. The IMF $B_Z$ distributions for the groups 1, 3 and 4 (all, foF2, and coherent foF2-$b_X$ intervals) are shown in Figure 14 a,b. While the distribution for the group 1 is almost symmetrical, both groups 3 and 4 are shifted to positive $B_Z$ values. This effect is stronger for the group 4. Group 4 relative occurrence was maximal at $3 < B_Z < 6$ nT (14b). The results for the SW speed are given in Figure 14 c,d. The main difference between the groups 1 and 3/4 was found in the bands centered at 500 and 300 km/s. The distributions for the latter groups are shifted to higher SW speeds. This might be due to an artifact of the initial selection procedure, which sets a lower boundary for the amplitude of geomagnetic pulsations. An important difference between the groups 3 and 4 is seen in Figure 14d, which demonstrates a maximal group 4 relative occurrence at high SW speed, namely in the band centered at 700 km/s.

Results for amplitudes of the SW dynamic pressure fluctuations are given at bottom panels (14 e,f). Here an increase of the most probable amplitude is clearly seen from group 1 to 3 and then to 4. This means that $P_{SW}$ amplitudes for coherent foF2-$b_X$ intervals are higher than that for the foF2 intervals, and for the foF2 intervals they are higher than that for all intervals. This effect is also seen for the group 4 for which relative occurrence reached 0.27 in the $\Delta P_{SW}$ range $(1.3 - 1.8)$ nPa against 0.13 at $\Delta P_{SW} < 0.9$ nPa.

To summarize, the foF2 fluctuations were preferably recorded under positive IMF $B_Z$, moderate $V \approx 500$ km/s SW speed, and the amplitudes of $P_{SW}$ fluctuations within $(0.6 - 0.9)$ nPa.

## 4  Discussion

The presented study of day-time fluctuations of foF2 in the $1 - 5$ mHz frequency range has been undertaken for quiet and moderately disturbed geomagnetic conditions when foF2 frequency can be unambiguously retrieved from ionograms. For that, a technique of automated scaling of foF2 was developed and verified by visual inspection.

Our results show a weak inter-relation between geomagnetic Pc5-6/Pi3 pulsations and foF2 variations in the same frequency range. The coherent foF2-$b_X$ pulsations can occur at typical Pc5 amplitudes. The most probable PSD band for these type of pulsations is recorded as often as at each 5-th interval. In these cases , the characteristics of magnetic pulsations such as the spectral content, polarization, and meridional PSD ratio allow to interpret them, following to (Baransky et al., 1995), as the Alfven FLR.

The Alfven FLR features are also seen in the spatial structure of Pc5 in the events 1 and 2 discussed as examples in Section 3.1.1. The Pc5 records at SOD and MAS for the event 1 are shown in Figure 15 together with meridional PSD ratio and phase difference. Waveforms of the pulsations are very similar, however, the phase is slightly delayed in MAS. The meridional PSD ratio is below 1 at low frequency flank of the spectrum, and it is growing at $f > 2.7$ mHz, reaching a maximum at 3.7 mHz.

(Figure 15b). The $f_2$ frequency of the second foF2-$b_X$ coherence maximum is near the frequency of maximal growth of meridional PSD ratio. A phase difference in this frequency band is about $-15°$ (Figure 15 c).

A similar result was obtained in the comprehensive statistical analysis of the correspondence between geomagnetic pulsations and pulsations in the Cosmic Noise Absorption (CNA) by Spanswick et al. (2005), who found that geomagnetic pulsations with FLR features demonstrate a better correspondence with CNA pulsations than non-FLR Pc5s. However, physical reasons for our and (Spanswick et al., 2005) results may be different, because of different energies of precipitating particle and types of geomagnetic pulsations. A detailed case study of pulsations in the magnetic field and the electron flux at four Cluster satellites located at different L-shells in the magnetosphere and geomagnetic and CNA pulsations on the ground (Motoba et al., 2013) showed rather complicated space distributions and time variations of geomagnetic and electron flux pulsations and their inter-relation. The pulsation in space was, probably, a mix of compressional and Alfven modes. The authors found that the amplitude of compressional mode was critical for effective modulation of electron flux, but the contribution of shear Alfven resonance was also non-negligible.

The foF2 was automatically retrieved from ionograms preferably near the noon and in the afternoon under moderately disturbed geomagnetic conditions. As a rule, the post-noon Pc5 are characterized by higher azimuthal wave numbers than morningside Pc5s (see Min et al. (2017) and references therein). They are often associated with kinetic modes originated from the wave-particle interactions (see e.g. Mager et al. (2013) and references therein). For these waves, the amplitudes on the ground are strongly attenuated by the ionosphere (Kokubun et al., 1989), while their amplitudes in the magnetosphere both in the magnetic field and in particle flux can be high (Baddeley et al., 2004). High-m waves generated by unstable ion distributions can effectively interact with ULF waves in Pc4 range (Baddeley et al., 2005). A comprehensive analysis of ion distribution functions in the magnetosphere undertaken by Baddeley et al. (2005) proved that the free energy of ion population provided observed magnitudes of high m ULF waves ionosphere.

These pulsations typically occur during auroral activations. Auroral substorms are followed by Pi3 pulsations (Kleimenova et al., 2002) and Pc5 waves with high and intermediate azimuthal wavenumbers (Zolotukhina et al., 2008; Mager et al., 2019). The substorm can generate magnetospheric waves with a wide spectrum of azimuthal wave numbers James et al. (2016). The large-scale waves are detected on Earth, whereas the small-scale waves modulate particle flux and are manifested in electron precipitation to the ionosphere. In such a situation, no strong dependence may be expected between the amplitudes of magnetic pulsations observed on ground and in the ionospheric electron density.

Geomagnetic pulsations recorded simultaneously in the magnetosphere and on Earth were studied by Watson et al. (2015). Their study clearly demonstrated that ULF waves effectively modulated electron flux at geostationary orbit and TEC in the ionosphere. Probably, also the flux of softer electrons, than those measured at GOES, was modulated. This allowed to explain observed values of TEC modulation. Different contribution of shear Alfven and compressional modes to the ULF power in the magnetosphere results in different TEC to magnetic field amplitude ratios, for the geomagnetic pulsations recorded on the ground.

This can explain the contrast in TEC to geomagnetic pulsations amplitude ratio found byWatson et al. (2015) and Pilipenko et al. (2014a) and between the two example events in the present study.

The foF2 to $b_X$ PSD ratio, $R_{F2-b}$, is essentially higher in the event 1, than in the event 2. Figure 16 shows a frequency dependence of $R_{F2-b}$ for these two events. The maximal difference is seen at about 2 mHz, and the difference is more than 2 orders of magnitude (more than an order of magnitude in the amplitude ratio). At frequencies of the second coherence maxima, the difference is more than an order of magnitude. The main visible difference between geomagnetic pulsations in these two events can be seen in their waveforms. The foF2 and geomagnetic variations during the event 1 have a well-correlated long-period part at $f < 1.6$ mHz. This may be a result of some global process, which is responsible for both geomagnetic pulsations and particle modulation. Fluctuations of SW dynamic pressure are one of the sources of global geomagnetic pulsations inside the magnetosphere (Kepko et al., 2002; Yagova et al., 2007; Viall et al., 2009). Indeed, the coherent foF2-$b_X$ pulsations are preferably associated with fluctuating $P_{SW}$, and periods of the foF2 fluctuations are often close to those of $P_{SW}$. However, in the present study we can't judge whether or not the ULF waves are global. To make such a conclusion a special study will be required with using satellite observations of the magnetic field and electron fluxes.

## 5   Conclusion

We suggested a technique for the automated detection of foF2 which allows to obtain data suitable for spectral estimates and comparison with Pc5-6/Pi3 geomagnetic pulsations. The foF2 variations show some inter-relation with Pc5-6/Pi3 geomagnetic pulsations, not only for extremely high, but for typical values of PSD, as well. Geomagnetic pulsations, for which foF2-$b_X$ coherence is higher at $f > 2.7$ mHz, i.e. near the Alfven resonant frequency at SOD, demonstrate properties of the Alfven FLR.

Variations of the ionospheric critical frequency foF2 were observed predominantly in the noon and afternoon during quiet or moderately disturbed geomagnetic conditions. The favorable conditions observing the foF2 fluctuations at the Pc5-6/Pi3 frequencies are the recovery phase of a weak or moderate geomagnetic storm, and moderate auroral activity at 6-hour maximal AE< 800 nT.

The favorable interplanetary conditions are a northward IMF, moderate $V$ (500 km/s), and amplitudes of SW dynamic pressure fluctuations about the 0.7 nPa.

The coherent foF2-$b_X$ pulsations tend to occur during an enhanced auroral activity (AE> 800 nT). Comparing to the non-coherent cases, they preferably occur under higher values of positive IMF $B_Z$, higher SW velocity, and larger amplitudes of SW dynamic pressure fluctuations.

*Sample availability.*   The foF2 values, obtained with Eq.(1) for all the intervals analyzed, are available both as jpeg figures and ASCII files. A file name has a structure SOD-YYYY-DDD-foF2, where YYYY is a year and DDD is a day number. Each ASCII file contains two columns:

1. time (seconds) from 00:00 UT

2. foF2 (MHz).

ASCII files with approximation parameters for the eq. 1. are also available, and a description of the columns is in the appr-readme.txt file.

*Author contributions.* N. Yagova: the approximation algorithm, cross-spectral and statistical analysis; A. Kozlovsky: pre-processing and visual check of ionosonde data; E. Fedorov: interpretation of results, analysis of wave parameters; O. Kozyreva: selection of events, algorithms and codes for visualization; all the authors: MS preparation

*Competing interests.* The authors have no competing interests

*Acknowledgements.* We thank SGO (http://www.sgo.fi/) for SOD magnetometer and ionosonde data, Finnish Meteorological Institute for
MAS magnetometer data, CDAWEB (https://cdaweb.gsfc.nasa.gov) for OMNI data, and World data center Kyoto (http://wdc.kugi.kyoto-u.ac.jp/index.html) for AE and Dst indices. The study was supported by the Academy of Finland grants 298578 and 310348 and RFBR grant 20-05-00787 (OK) and state contract with IPE (NY, EF). Useful discussions with V.A. Pilipenko are appreciated. The authors thank the topical editor and both referees for helpful remarks.

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

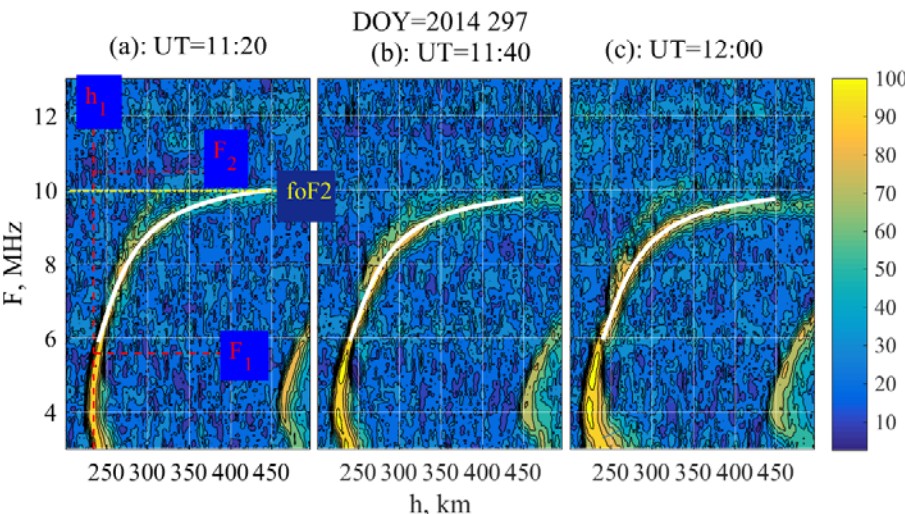

**Figure 1.** Examples ionograms and approximations of F(h) dependence with eq. (1). foF2 and $h_1, F_1, F_2$ approximation parameters are shown at panel 1(a). Reflection intensity is shown in color in dB.

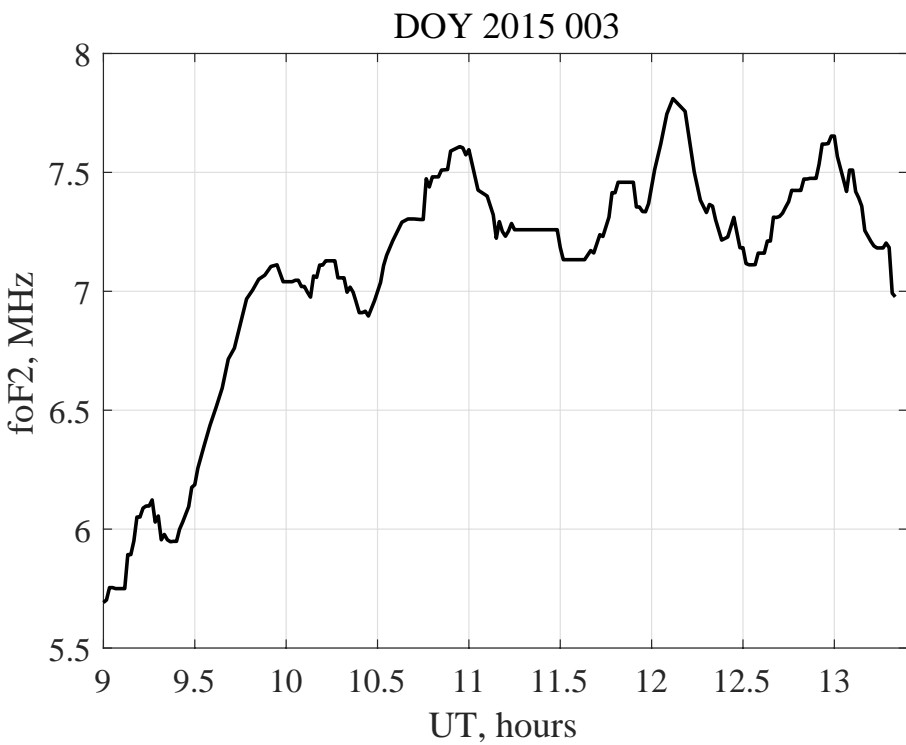

**Figure 2.** Variation of foF2 frequency during 4.5 hours on day 2015 003 , obtained with eq. (1)

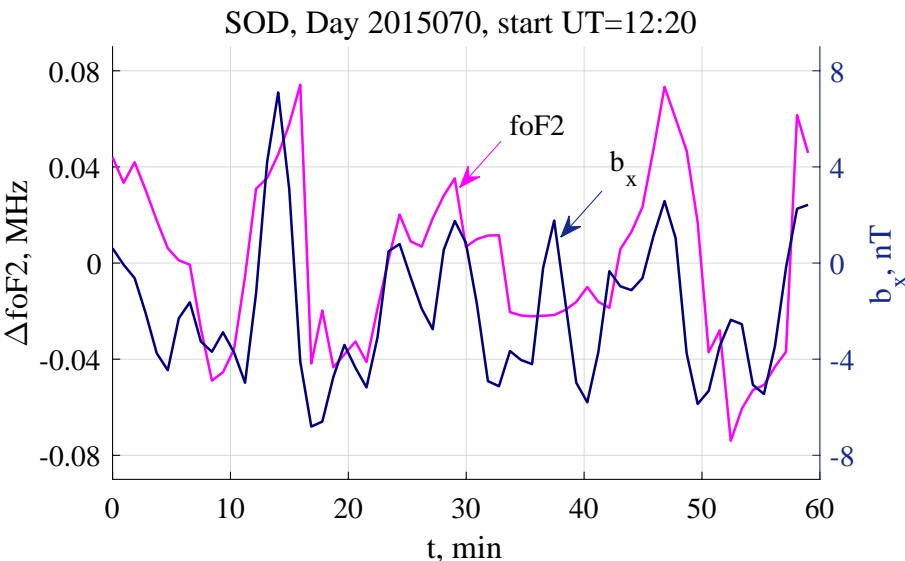

**Figure 3.** Variation of foF2 and geomagnetic pulsations at SOD during event 1

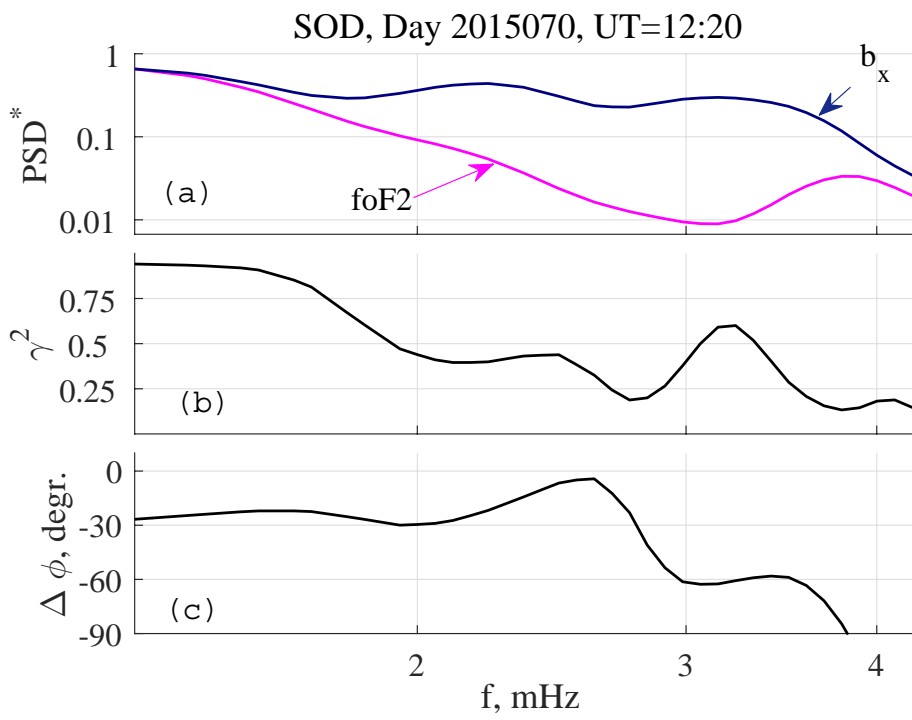

**Figure 4.** Spectral parameters for the event 1: (a) normalized PSD spectra of foF2 and $b_X$ pulsations, (b) spectral coherence between foF2 and $b_X$; (c) phase difference

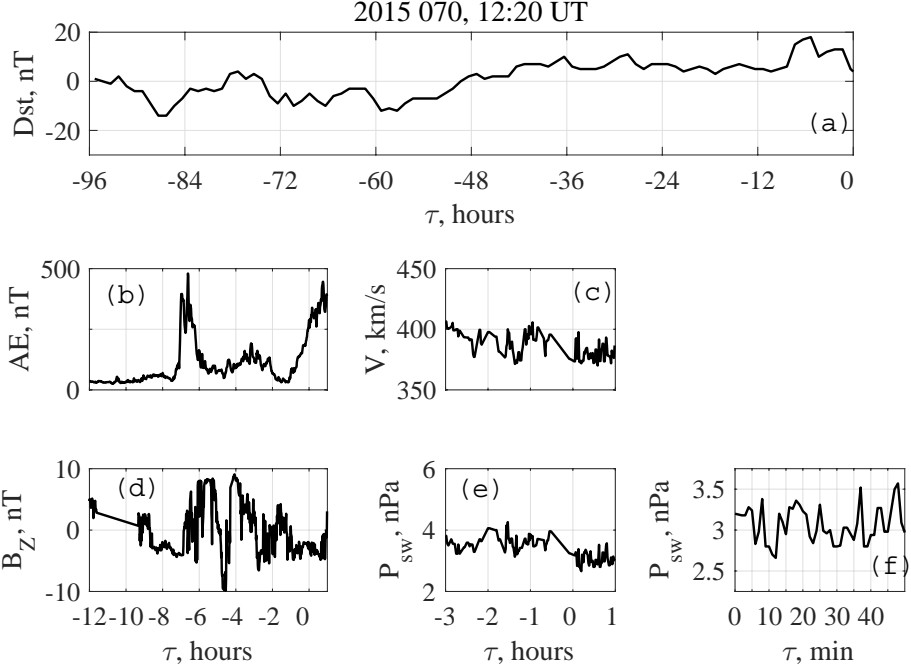

**Figure 5.** Space weather conditions for the event 1. (a) Dst index during last four days; (b) AE index during the interval and 12 hours before; (c) SW speed during the interval and 3 hours before; (d) IMF $B_Z$ during the interval and 12 hours before; (e) SW dynamic pressure during the interval and 3 hours before; (f) details of SW dynamic pressure fluctuations during the interval.

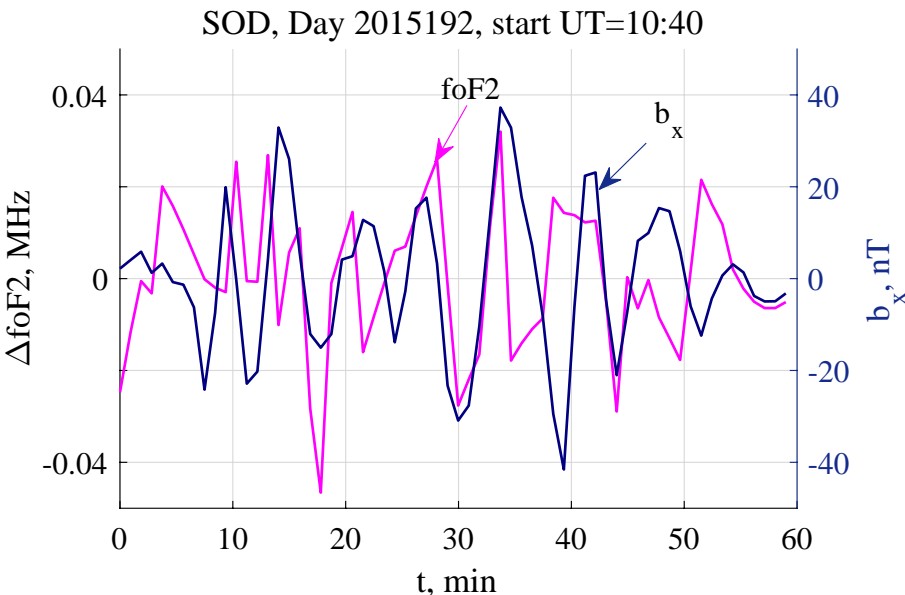

**Figure 6.** Variation of foF2 and geomagnetic pulsations at SOD during event 2

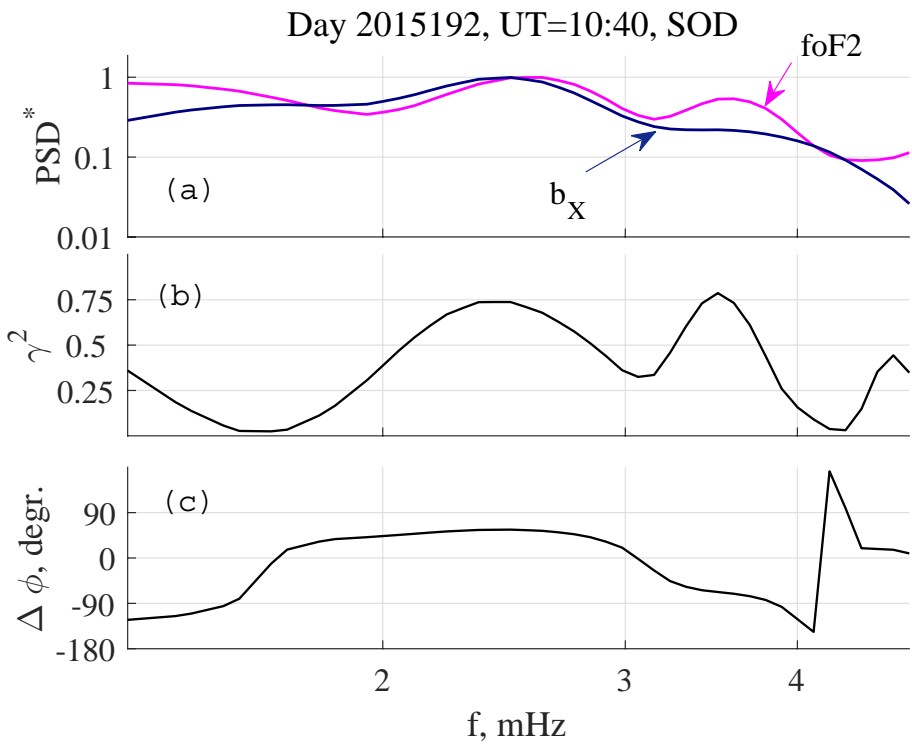

**Figure 7.** Spectral parameters for the event 2: (a) normalized PSD spectra of foF2 and $b_X$ pulsations, (b) spectral coherence between foF2 and $b_X$; (c) phase difference.

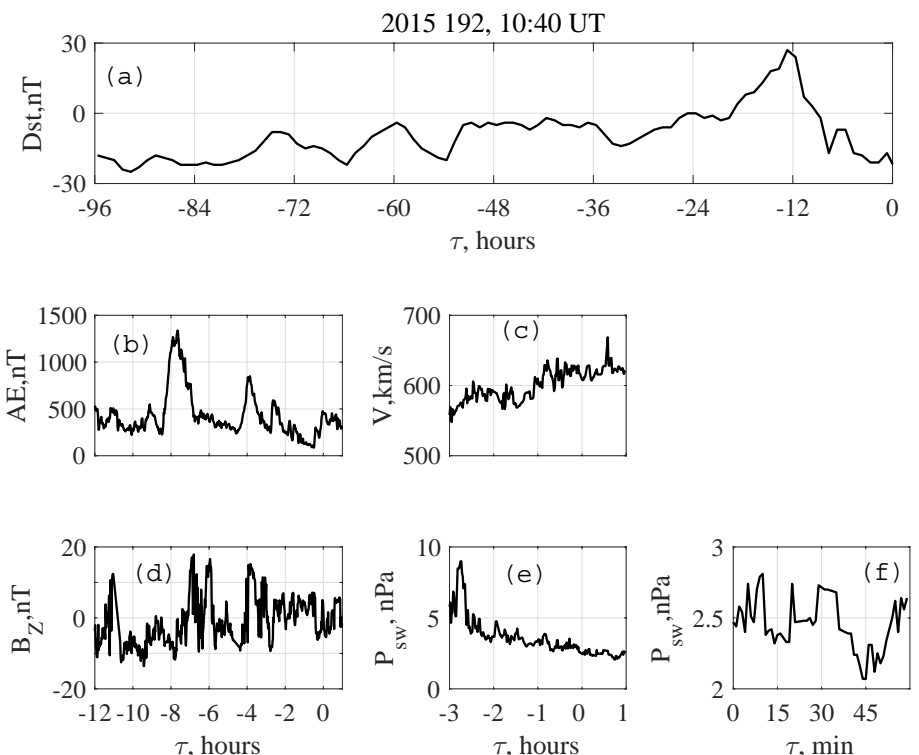

**Figure 8.** Space weather conditions for the event 2. (a) Dst index during last four days; (b) AE index during the interval and 12 hours before; (c) SW speed during the interval and 3 hours before; (d) IMF $B_Z$ during the interval and 12 hours before; (e) SW dynamic pressure during the interval and 3 hours before; (f) details of SW dynamic pressure fluctuations.

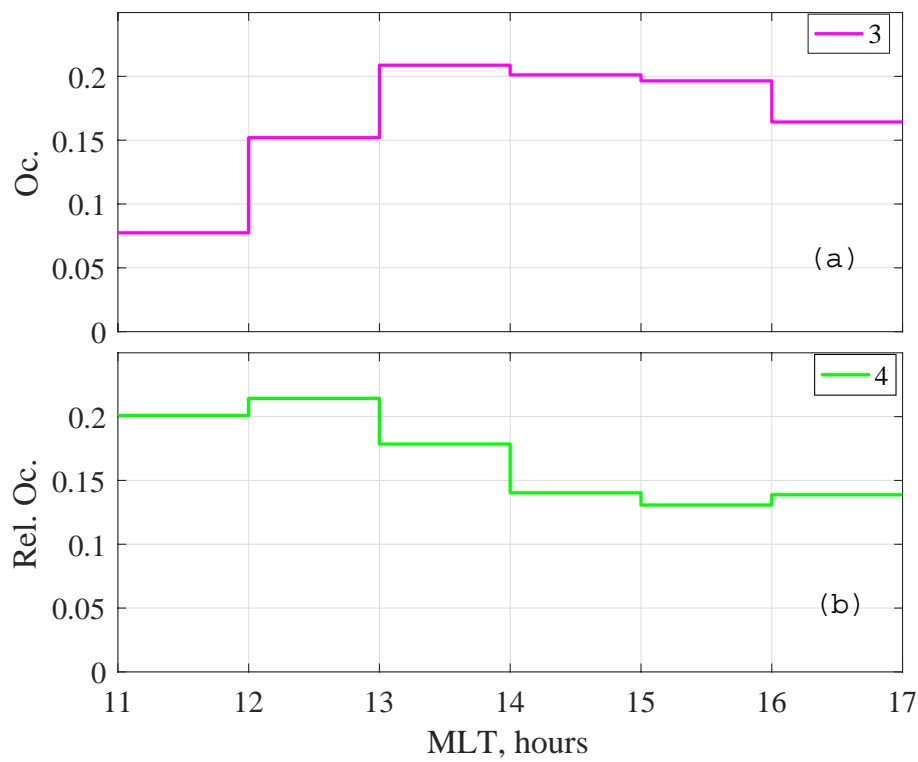

**Figure 9.** (a): The average MLT distribution of occurrence of foF2 intervals (group 3). (b): Relative occurrence of coherent foF2-$b_X$ pulsations (group 4) for $\gamma^2 > 0.5$

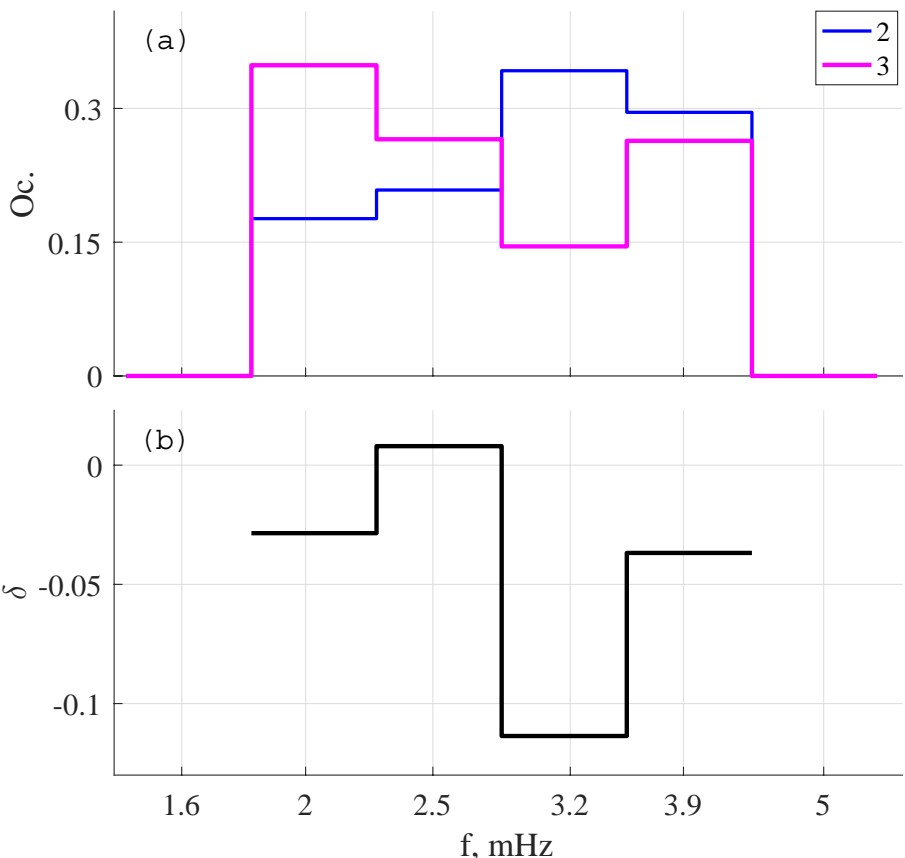

**Figure 10.** Frequency histogram, for $b_X$ (group 2) and foF2 (group 3) fluctuations (a) and for the parameter $\delta = \log(\Delta_{f2,0}/\Delta_{f2,R})$, where $\Delta_{f2,0}$ and $\Delta_{f2,R}$ are the mean values of $\Delta_{f2}$ for simultaneous and random intervals (b).

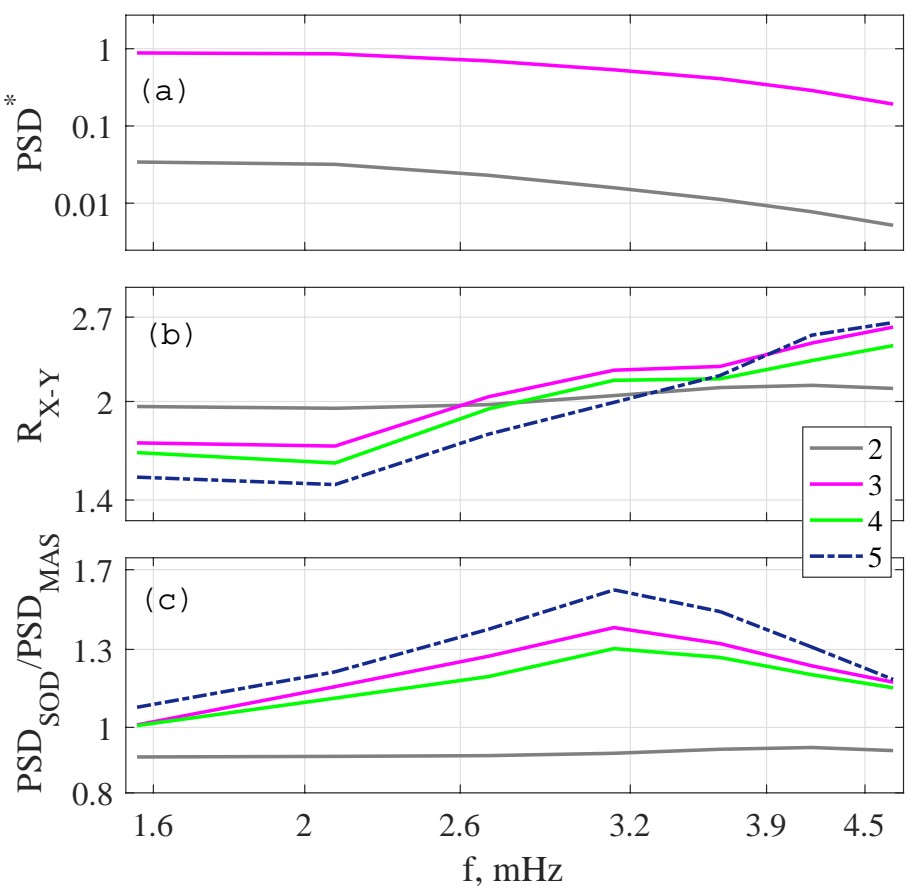

**Figure 11.** Comparison of Pc5-3/Pi3 parameters for Pc5-3/Pi3 (group 2), foF2 (group 3), coherent foF2-$b_X$ (group 4), and high frequency coherent foF2-$b_X$ (group 5)intervals: (a) PSD; (b) $R_{X-Y} = \mathrm{PSD}_X/\mathrm{PSD}_Y$; (c) meridional PSD ratio $\mathrm{PSD}_{\mathrm{SOD}}/\mathrm{PSD}_{\mathrm{MAS}}$

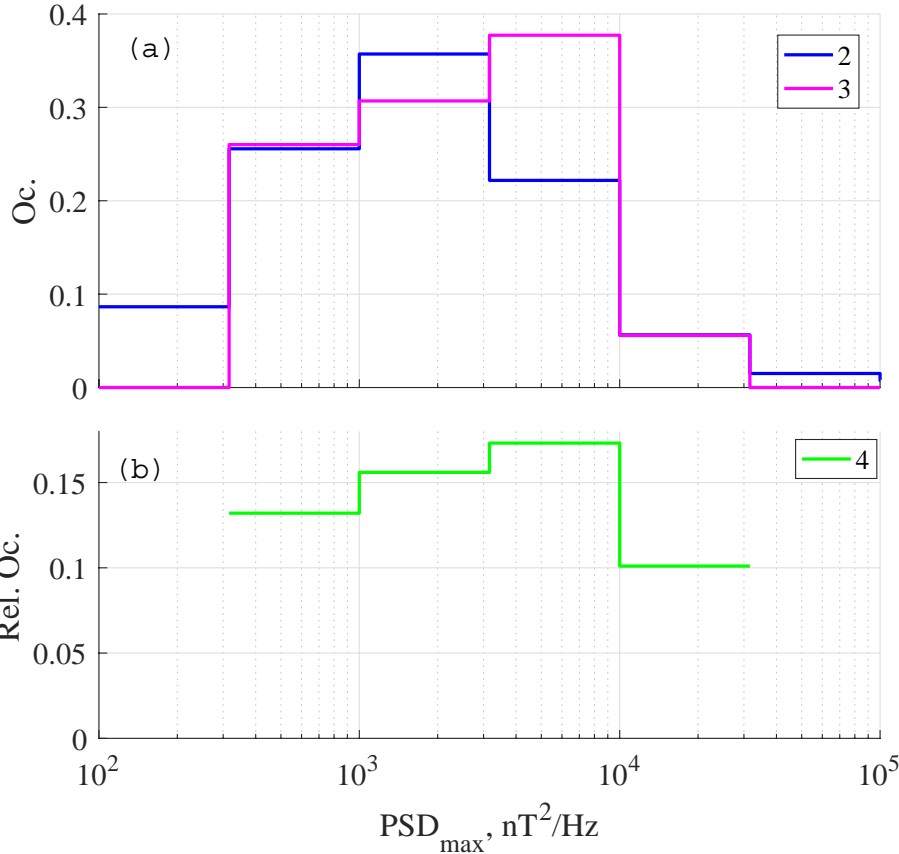

**Figure 12.** PSD distributions for groups 2 and 3 (a) and group 4 relative occurrence (b).

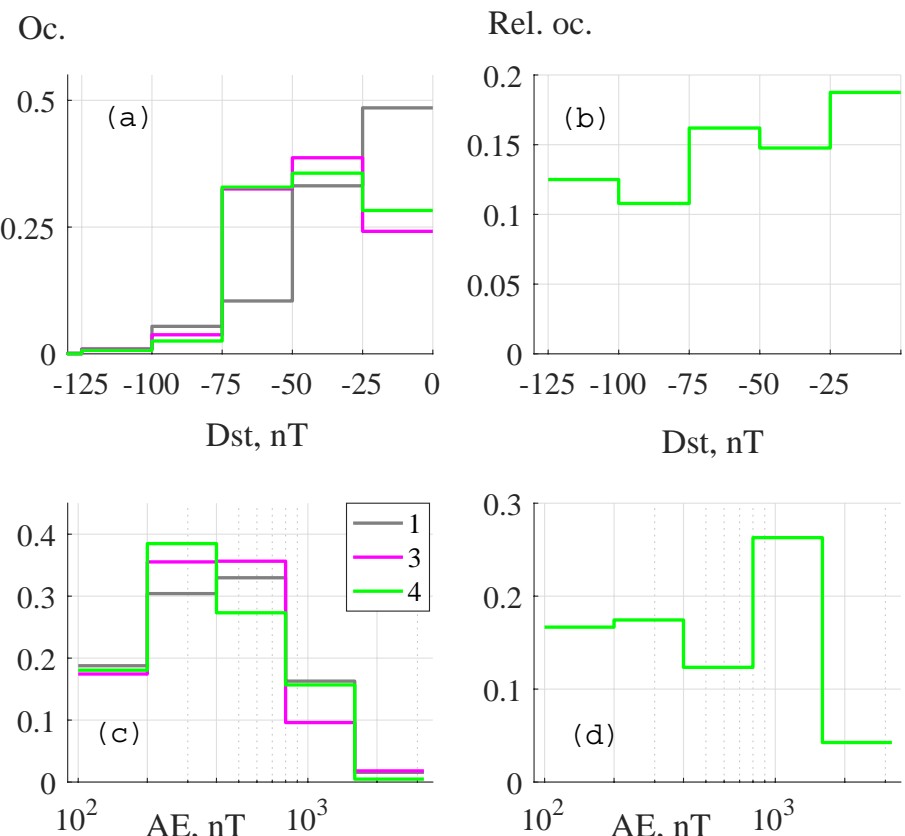

**Figure 13.** Left. Activity indices distributions for the groups 1, 3, and 4: Dst (a) and AE (c). Right. The same for the group 4 relative occurrence: Dst (b) and AE (d)

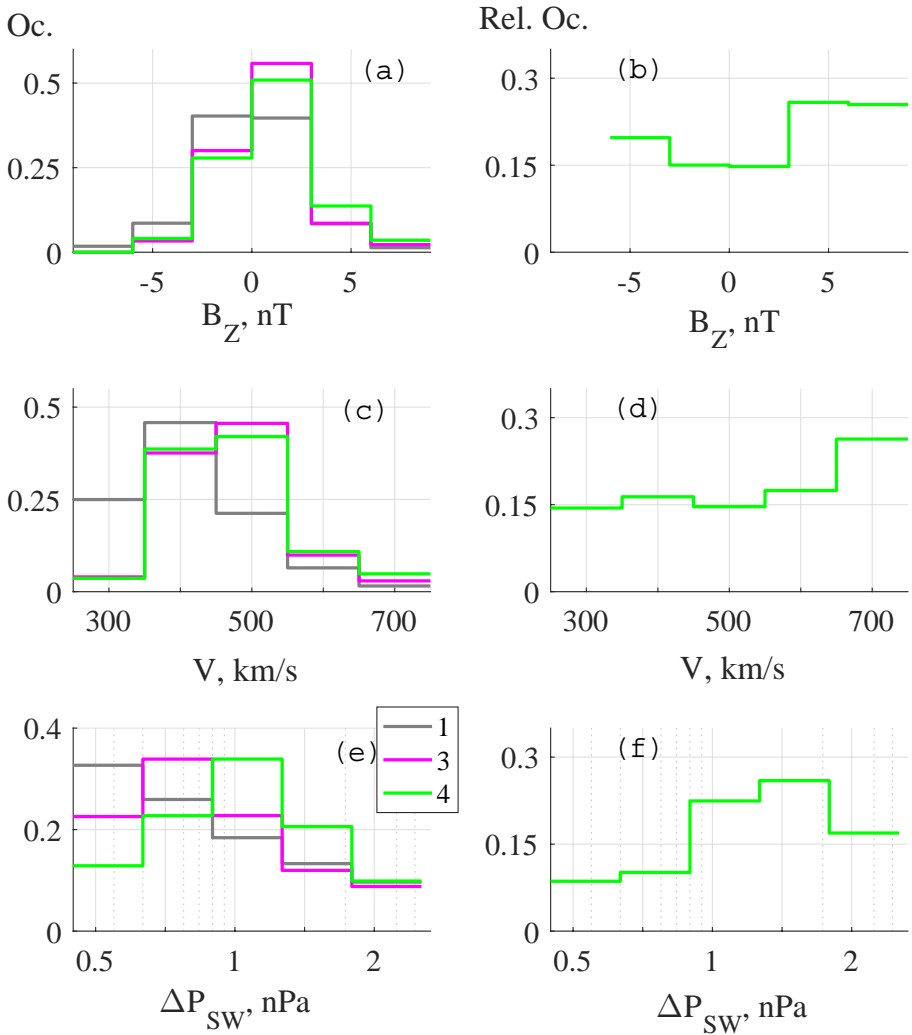

**Figure 14.** Left. SW/IMF parameter distributions for the groups 1, 3, and 4: IMF $B_Z$ (a), $V$ (c), and $\Delta P_{SW}$ (e). Right. The same for the group 4 relative occurrence: $B_Z$ (b), $V$ (d), and $\Delta P_{SW}$ (f).

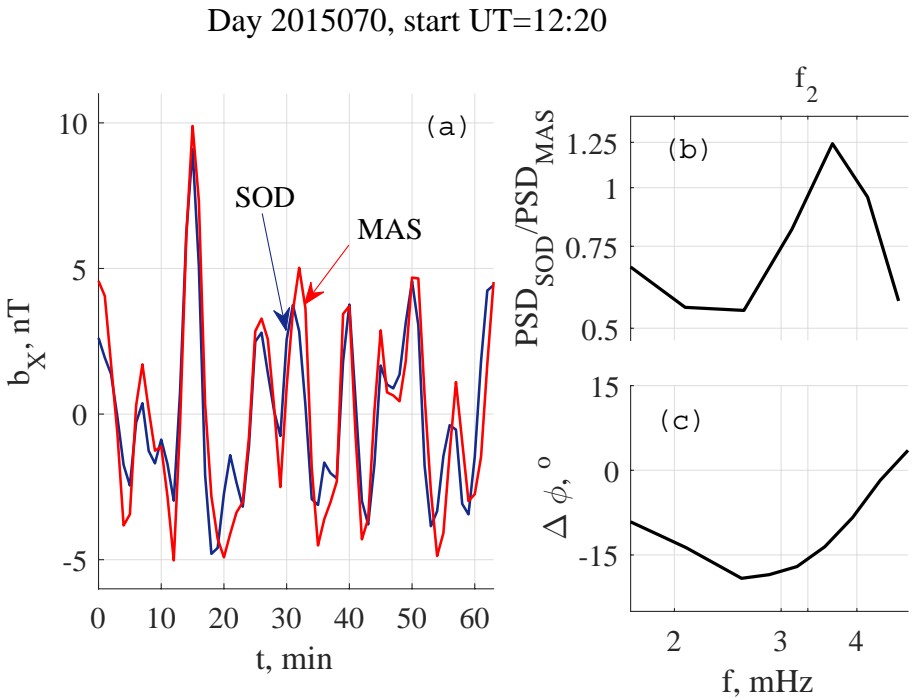

**Figure 15.** Illustration of FLR signatures of geomagnetic pulsations on 11 March (event 1). Pulsation waveforms at SOD and MAS (a); meridional PSD ratio (b) and phase difference (c).

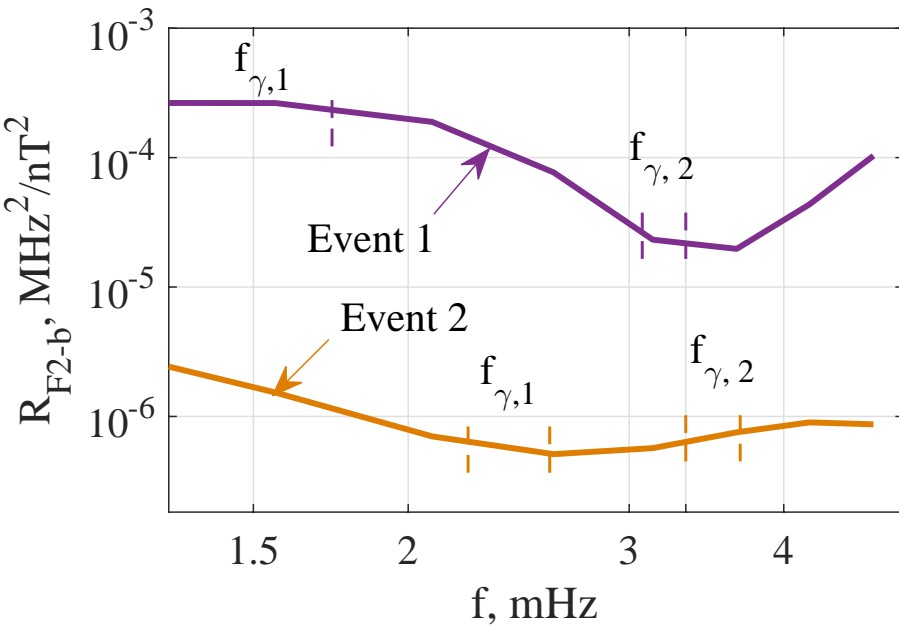

**Figure 16.** Comparison of foF2 to $b_X$ PSD ratios $R_{F2-b}$ for the events 1 and 2. The bands of high foF2-$b_X$ coherence are shown with vertical dash lines

**Table 1.** Coordinates and other parameters of IMAGE stations

| Station | Geographic | | CGM | | L | MLT |
|---------|------------|------|------|-------|------|----------|
| | LAT | LON | $\Phi$ | $\Lambda$ | | midnight |
| SOD | 67.37 | 26.63 | 64.2 | 106.5 | 5.37 | 21:12 |
| MAS | 69.46 | 23.70 | 66.5 | 105.5 | 6.37 | 21:18 |

Corrected geomagnetic (CGM) latitude $\Phi$ and longitude $\Lambda$, apex of the magnetic field line $L$,

and UT of magnetic local midnight are calculated online with

http://omniweb.gsfc.nasa.gov/vitmo/cgm.html

**Table 2.** Intervals with foF2 obtained from Eq. (1) checked visually. Years 2014-2015.

| Year | Month | Day | DOY | Start UT | Final UT | | Year | Month | Day | DOY | Start UT | Final UT |
|------|-------|-----|-----|----------|----------|--|------|-------|-----|-----|----------|----------|
| 2014 | 2 | 24 | 55 | 9:00 | 14:59 | | 2015 | 1 | 3 | 3 | 9:00 | 13:20 |
| 2014 | 4 | 11 | 101 | 8:00 | 15:59 | | 2015 | 1 | 22 | 22 | 9:00 | 12:10 |
| 2014 | 5 | 24 | 144 | 7:00 | 15:00 | | 2015 | 1 | 30 | 30 | 9:00 | 12:20 |
| 2014 | 6 | 8 | 159 | 7:00 | 15:59 | | 2015 | 2 | 15 | 46 | 9:30 | 13:10 |
| 2014 | 6 | 21 | 172 | 9:00 | 15:59 | | 2015 | 3 | 8 | 67 | 9:00 | 13:40 |
| 2014 | 8 | 18 | 230 | 8:00 | 15:59 | | 2015 | 3 | 10 | 69 | 9:00 | 15:59 |
| 2014 | 9 | 13 | 256 | 8:00 | 15:59 | | 2015 | 3 | 11 | 70 | 10:50 | 15:00 |
| 2014 | 9 | 14 | 257 | 8:00 | 15:59 | | 2015 | 4 | 4 | 94 | 9:00 | 14:20 |
| 2014 | 9 | 15 | 258 | 8:00 | 15:59 | | 2015 | 5 | 14 | 134 | 9:00 | 14:20 |
| 2014 | 9 | 16 | 259 | 8:00 | 15:59 | | 2015 | 5 | 15 | 135 | 9:10 | 15:59 |
| 2014 | 9 | 25 | 268 | 8:00 | 15:59 | | 2015 | 5 | 19 | 139 | 8:40 | 14:50 |
| 2014 | 10 | 15 | 288 | 8:00 | 15:10 | | 2015 | 5 | 20 | 140 | 8:00 | 11:20 |
| 2014 | 10 | 19 | 292 | 8:00 | 14:30 | | 2015 | 5 | 20 | 140 | 12:10 | 15:59 |
| 2014 | 10 | 24 | 297 | 8:00 | 12:40 | | 2015 | 6 | 12 | 163 | 7:20 | 15:59 |
| 2014 | 10 | 31 | 304 | 8:00 | 14:59 | | 2015 | 6 | 30 | 181 | 7:00 | 8:40 |
| 2014 | 11 | 6 | 310 | 8:00 | 14:59 | | 2015 | 6 | 30 | 181 | 9:10 | 12:20 |
| 2014 | 11 | 7 | 311 | 8:00 | 14:59 | | 2015 | 6 | 30 | 181 | 13:30 | 15:59 |
| 2014 | 11 | 8 | 312 | 8:00 | 13:30 | | 2015 | 7 | 11 | 192 | 12:20 | 15:59 |
| 2014 | 11 | 10 | 314 | 9:00 | 12:30 | | 2015 | 7 | 12 | 193 | 7:20 | 15:30 |
| 2014 | 11 | 11 | 315 | 9:00 | 14:40 | | 2015 | 7 | 15 | 196 | 8:20 | 10:30 |
| 2014 | 11 | 12 | 316 | 9:00 | 14:59 | | 2015 | 7 | 15 | 196 | 11:10 | 15:59 |
| 2014 | 12 | 7 | 341 | 9:00 | 12:20 | | 2015 | 7 | 21 | 202 | 8:00 | 11:10 |
| 2014 | 12 | 10 | 344 | 9:00 | 14:59 | | 2015 | 7 | 21 | 202 | 12:50 | 15:59 |
| 2014 | 12 | 13 | 347 | 9:00 | 14:00 | | 2015 | 9 | 8 | 251 | 8:00 | 12:20 |
| 2014 | 12 | 27 | 361 | 9:00 | 14:20 | | 2015 | 9 | 8 | 251 | 12:50 | 15:20 |
| 2014 | 12 | 29 | 363 | 9:00 | 12:30 | | 2015 | 9 | 16 | 259 | 8:00 | 15:59 |
| | | | | | | | 2015 | 9 | 22 | 265 | 8:00 | 15:59 |
| | | | | | | | 2015 | 9 | 23 | 266 | 8:00 | 15:59 |
| | | | | | | | 2015 | 12 | 25 | 359 | 9:00 | 10:50 |