# Peer review of "Even moderate geomagnetic pulsations can cause fluctuations of foF2 frequency of the auroral ionosphere"

_Annales Geophysicae, 2020_

## Referee Comment (RC1) · Anonymous Referee #1 · 29 May 2020

Title: Even moderate geomagnetic pulsations can cause fluctuations of foF2 frequency of the auroral ionosphere.

The authors investigate the relation between colocated, simultaneous fluctuations in the F2 critical frequency and geomagnetic time series. They developed an automated method for inferring foF2 frequency from the ionograms. Events with foF2 frequencies modulated in the Pc5/Pi3 frequency range are analysed. The properties of a subset of events with coherence greater than 0.5 are compared to the average properties of the whole population of the events. The authors found that coherent events favour moder-

ate geomagnetic conditions and show typical features of field line resonances. At the same time, it is noted that the automated detection of foF2 is not applicable to disturbed conditions. The paper, in general, is difficult to read and follow mainly because of its poor language. We strongly recommend the authors to use some spell-check tool to improve the quality of the presentation of their thoughts. A figure illustrating the automated detection of the critical frequencies would be helpful for the reader. The focus of the paper is on coherent events, however, neither the 'event' is exactly defined, nor the significance level of the calculated coherence is given. There is also some inconsistency in the paper about at what frequency the coherence is taken (f1 vs f2: statistics in Fig 10 vs. case studies ). Results presented in Fig 11 again suggest a link to f2 (at least based on the case studies). The relative occurrence of coherent events is very low ($\sim$ 3%). The statistics support that coherent events tend to occur under moderately disturbed geomagnetic and interplanetary conditions. However, the significance of this result is not clear due to 1. the low relative occurrence of coherent events, 2. the unknown significance level of coherence, 3. the limitations of foF2 detection under disturbed conditions, 4. the applied normalisation on which limited information is given. It was also not investigated how often the coherent events show up provided the conditions are favourable (moderate disturbance). Without this information the study is not complete and cannot be judged. I recommend a major revision. Below I give a list of my minor comments in two groups. The first group relates to science, the second to the language. The latter is far from being complete. It would have been a long list.

Minor comments on the scientific content

l 97: "about 10 nT and 0.08 MHz": Revise these values based on Fig 3!

l 100 what is the significance level for the coherence values in these calculations?

l 112: "about 80 nT and 0.08 MHz": although geomagnetic variations are several times greater here then for event 1, the foF2 variations are smaller. Comment?

l 126: A MLT distribution of occurrence of the foF2 variations –> The MLT distribution

of the occurrence of foF2 variations Under what conditions? What criteria define an event?

l 128: "frequency distributions of geomagnetic and foF2 pulsations": in general? I guess the distribution is based geomagnetic pulsation events simultaneous with foF2 events.

Figure 9: What is D (vertical axis)? Relative occurrence

Figure 10: Why the distribution of the first spectral peaks is presented. In your example events f2 has the higher coherence and corresponding Psw fluctuations. Are not your examples presented typical for the coupling between foF2 and geomagnetic variations?

l 131: "spectral coherence at SOD" : at what frequency? coherence at f1?

l 132: Give the significance level!

Figure 10 b) Mark the significance level in this plot!

l 142: some information on the derivation of the weight functions and how they applied to normalize the data is needed

l 153: "the 4-day minimum Dst and 6-hour maximal AE": intervals centred or preceding the coherent event?

l 158-162 Do coherent events occur under severely disturbed conditions, just they cannot be observed? Or they do not occur under those conditions at all? How does this observational limitation affect your conclusions?

l 165: "3-hour mean values of BZ and V and 3-hour maximal value of $\Delta$Psw": in which interval? (same issue as above)

l 181: "Amplitude of SW dynamic pressure fluctuations show an association with occurrence of coherent foF2 − B pulsations": only 2 examples were presented in favour of this statement. Figure 13 b) does not yield any information on the spectral content of

the pressure variations, and hence their relation to the coherent frequency. You seem to focus your statistics on f1 (first peak, e.g. Fig 10 a; coherence at f1), while your 2 examples had their relation with the SW pressure variations at f2.

l 185: refer to your observations relevant presented in Fig 11. and show how they support the FLR nature of the coherent subset

l 193: "The picture changed dramatically": be more specific!

Further comments:

l 1: "variations of the critical frequency": maybe "modulation" of the critical frequency could also be used here

l 1: o-mode radiowave –> o-mode radio waves

l 2: in 1–5 mHz –> in the 1–5 mHz

l 4: delete "daytime Pc5/Pi3 geomagnetic pulsations and" [foF2 is obviously not detected in geomagnetic pulsations]

l 6: at SOD station –> observed at SOD station

l 6: with the data of a station pair located at the same magnetic meridian –> using the data of a station pair located along the same magnetic meridian

l 8: Meanwhile, –> "At the same time," OR "However,"

l 8: "the analysis of geomagnetic and foF2 variations show intervals with noticeable coherence for both horizontal components" –> "the analysis of geomagnetic ad foF2 variations shows intervals of significant [OR remarkable] coherence with both horizontal geomagnetic components" [foF2 does not have any components]

l 11: averaged –> the average

l 11: coherent to –> coherent with

l 13: show –> shows

l 14: show –> shows [I suggest to use some synonym of 'show', such as 'reveal', 'indicate'. Use an online Thesaurus for finding synonyms]

l 19: Majority of publications are based on the radar observation –> Majority of publications on the topic are based on radar observations

l 20: of electron concentration at certain altitude –> of the electron concentration at a certain altitude

l 23: with mainly compressional mode of MHD wave in the magnetosphere –> with mainly compressional mode magnetospheric waves

l 26: An effect of TEC modulation by ULF wave –> The effect of TEC modulation by a ULF waves

l 27: and zones –> and also from zones

l 28: observed pulsations –> the observed pulsations [a large number of articles are missing from the text, check!]

l 31: the recovery phase of the magnetic storm –> the recovery phase of a magnetic storm

l 33: aimed on variations –> aimed at comparing variations

l 37: It makes an ionogram –> It obtains an ionogram recording

l 40: 10 s sampling rate –> 10 s sampling period/interval

l 40: and we also use the data of the MAS station, which is a part of IMAGE –> we also use data of recorded at MAS station of the IMAGE network

l 42: To analyzed –> To analyze

l 43: and also Dst and AE indexes are used –> as well as Dst and AE indexes

l 46: with quality and time resolution enough –> with good quality and time resolution is enough

l 56: for the reader's sake refer to your Fig 1 here.

l 58: Lorentsian –> Lorentzian

l 60: 235 km –> 235 km.

l 60: Coefficients f1, $\Delta f = f2 - f1$, k, and $\alpha$ are found as a result of fitting procedure, described below. –> A fitting procedure described below is used to find f1, $\Delta f = f2 - f1$, k, and $\alpha$. [f1,$\Delta f$, $\alpha$ are not coefficients] What are the meaning of f1 (I guess f at h1) and f2?

l 61: boundary is determined as a line –> boundary consists of a set of (h,f) points

l 62: Signal intensity I at the boundary should be high –> Signal intensity I is high

l 63: Amplitude ratio R of the signal intensity at the boundary line to the power above it should also be high –> The contrast between the peak and the background (characterized by the amplitude ratio R) is high [or similar, your version is confusing. Intensity to power ratio called amplitude ratio... It is not clear what is 'above'. At higher frequency?]

l 64: As four fitting factors are used –> We then fit Eq(1) to the detected boundary points. As four fitting factors are used

l 64: organized and a parameter –> organized. A parameter

l 65: over the "cross" in space of parameters –> over the parameter space [?]

l 65: where x is a point in the space of parameters, and i is a parameter number –> where 'x' is a point in the parameter space, and 'i' identifies the parameter [and what is c? ]

l 64: Give a representative example, e.g. the values of the parameters used to derive the fits presented in Fig 1!

l 68: time dependence f(t): Do you mean the time dependence foF2(t)?

l 69: give a typical value of t1!

l 71: the other [??? or another]

l 73: Examples of approximation curves are given in Figure 1:

Figure 1: Complete the figure caption by including "the fitted curves are plotted over the ionograms in yellow" or similar. Add a reference to the fitting curves in Fig 1 in the main text, as well.

l 76: pictures –> plots

l 79: Note, that the ionograms are rotated by 90âŮẹ in respect to usual $f - H$ presentation: This sentence should come earlier! (with respect to)

l 80: foF2 –> foF2 values

l 84: Statistical analysis: Statistical analysis of what?

l 84: interval –> intervals

l 85: We studied –> We studied the effect/influence of .... on...

l 87: resolution, enough –> resolution high enough

l 90: Cross-spectra are calculated for foF2 variations, on one hand, and components of the geomagnetic field pulsations, on the other hand. –> Cross-spectra are calculated between foF2 variations and components of the geomagnetic field pulsations.

l 100: "at low frequency part of spectrum f < 2 mHz" –> "in the low frequency part (f < 2 mHz) of spectrum"

l 101: peak with maximal y2 = 0.6 –> peak with y2 = 0.6

l 129: "with frequencies (f1 > 3.7 mHz)" –> "with frequencies above 3.7 mHz

l 130: "The distribution of Pc5/Pi3 intervals over foF2 − b spectral coherence at SOD are shown in Figure 10b for two" –> "The histogram of the foF2 − b spectral coherence at SOD is shown in Figure 10b for the two"

l 137: "a question arises about the pulsation properties and external parameters, favorable for their occurrence": rephrase!

l 138: "the geomagnetic pulsations" –> "a subset of the geomagnetic pulsations"

l 139: " with all the intervals, selected" –> " with all the events selected"

l 142: "calculated with the weight functions, which are found from" –> "calculated with weight functions derived from"

l 143: "coherent and pulsations and averaged" –> "coherent pulsations and averaged"

l 151: indexes –> indices

l 159: "limited by" –> "limited to"

l 186: "in coherent foF2 − Bx pulsations" : delete. This information is already given earlier in the sentence.

l 197: "For the first time, a statistical study of foF2 variations in Pc5/Pi3 range and their relation to geomagnetic pulsation in the conjugated position at SOD station and its spatial distribution along a magnetic meridian." Check the sentence (missing predicate).

---

## Referee Comment (RC2) · Anonymous Referee #2 · 19 Aug 2020

In their manuscript "Even moderate geomagnetic pulsations can cause fluctuations of foF2 frequency of the auroral ionosphere", Yagova et al. explore variations of the ionosphere F2 region critical frequency (foF2) and ultra-low frequency (ULF) waves in the Pc5 and Pi3 frequency bands detected at auroral latitudes.

Using ground magnetometer and ionosonde data spanning years 2014 and 2015, the authors examine the power, coherence and phase difference of perturbations in the daytime ionosphere and Pc5/Pi3 geomagnetic pulsations, distinguishing a subset of events during periods of magnetic quiescence and moderate magnetic storms with coherence greater than 0.5 from ULF wave signatures in the ionosphere observed under

conditions favourable to strong geomagnetic storms. This extends previous studies by Pilipenko et al. (2014a and 2014b) that considered ULF wave-driven oscillations in the ionosphere F2 region during strong and small magnetic storms.

Furthermore, the manuscript presents a new methodology to the automated detection of the foF2 critical frequency from ionograms that could be of interest for the research community working on determining factors that influence the amplitude and phase of perturbations in the ionosphere as these are detected on the ground. There are, however, several issues that hinder my recommendation of this manuscript for publication in Annales Geophysicae in its present form.

There are major issues with the English language use, several typographical errors and in general, it is poorly written making it difficult to understand the scientific rationale behind this study. For example, in line 19, it reads: "Modulation of ionospheric parameters by Pc5 pulsations was reported . . .", without detailing which parameters are meant here. In the same line, it goes on to say: "Majority of publications are based on the radar observation . . ." (which would more correctly read "The majority of publications are based on radar observations . . ."), without making it clear to which publications the authors refer.

It would be worthwhile to establish in the Introduction the need for a study such as the present by listing past publications focused on perturbations in the ionosphere driven by ULF waves. Early results on geomagnetic pulsations in the ULF wave frequency range associated with total electron content (TEC) fluctuations date back to 1976 and include the following:

- Davies & Hartmann (1976), Short-period fluctuations in total columnar electron content, Journal of Geophysical Research, https://doi.org/10.1029/JA081i019p03431

- Okuzawa & Davies (1981), Pulsations in total columnar electron content, Journal of Geophysical Research, https://doi.org/10.1029/JA086iA03p01355

Total electron content variations have been proven a powerful tool in the detection of ionospheric signatures of ULF waves at high latitudes as well as data from ionosondes exploiting the radio-wave reflecting properties of the ionosphere, as it is detailed by Watson et al. (2015). It is not clear to me and perhaps the reader how the results of Watson are different from those of Kozyreva et al. (2019) briefly mentioned in line 29. Nor the difference with those of Pilipenko et al. (2014b) derived from data collected during a different magnetic storm.

The following publications could be added to improve the placement of this work in the context of existing literature:

- Baddeley et al. (2005), On the coupling between unstable magnetospheric particle populations and resonant high-m ULF wave signatures in the ionosphere, Annales Geophysicae, https://doi.org/10.5194/angeo-23-567-2005

- Buchert et al. (1999), Ionospheric conductivity modulation in ULF pulsations, Journal of Geophysical Research, https://doi.org/10.1029/1998JA900180

In lines 31 and 32, the authors note that the association of waves with moderate amplitudes with variations of the foF2 critical frequency have not been studied. However, how their amplitude is defined as moderate is not described nor later in the manuscript. As mentioned in the title of the manuscript, the reader is waiting for more details on these moderate geomagnetic pulsations, in my mind.

In lines 62 and 63, could the authors explain in quantitative terms how high the signal intensity at the reflection boundary should be as well as the amplitude ratio of the signal intensity at the reflection boundary to the power above it?

Later, in lines 68 and 71, the authors note that a threshold for the time derivative of the foF2 critical frequency is calculated from the variance over a time interval of length t1. Is the variance of the foF2 critical frequency meant? How is the length of the time interval t1 defined?

Section 2.2 would benefit from an ionogram on which the described method has been used to detect the ionosphere F2 region critical frequency, clearly illustrating the new method for the foF2 critical frequency automated detection.

In Figures 4 and 7, it would be worthwhile to note the frequency of the primary and secondary maximum in power and provide further explanation at which frequency the coherence is taken for the statistics provided in Section 3.1.2.

In Section 3.1.1, in addition to the details offered for the two intervals in March and July 2015, the two examples could be utilised to introduce the criteria set for selecting similar events for subsequent statistical analysis.

In Figures 9, 10, 12 and 13, as these are described in Section 3.1.2, what does "occurrence" and the symbol "D" mean in this context? Do the authors refer to "probability of occurrence"?

As they stand, the conclusions reached and briefly summarised in the first paragraph of Section 4 of this manuscript are a bit vague. Although it is suggested that this study is focused on variations of the ionosphere's critical frequency foF2 during quiet and moderately disturbed geomagnetic conditions, the most favourable values of the Dst index lay between -100 and -50 nT. Under such conditions, how often would it be expected to detect events are associated with ULF geomagnetic pulsations? How would the low occurrence rate (3%) of coherent events change if periods of highly disturbed conditions or quiescence were excluded? Please also consider commenting on the solar wind conditions that are favourable for the occurrence of coherent events and specifically, provide the range of solar wind speed and dynamic pressure values.

Lastly, there are inconsistencies in the referencing style and specifically, on page 9 and 10, the year of publication in Mager et al. (2013), Min et al. (2017) and Viall et al. (2009) should be moved to the end of each reference.

---

## Author Comment (AC1) · 14 Sep 2020

Dear Referee,

thank you for your helpful report. Point-by-point answers are given below

Anonymous Referee #2 In their manuscript "Even moderate geomagnetic pulsations can cause fluctuations of foF2 frequency of the auroral ionosphere", Yagova et al. explore variations of the ionosphere F2 region critical frequency (foF2) and ultra-low frequency (ULF) waves in the Pc5 and Pi3 frequency bands detected at auroral latitudes. Using ground magnetometer and

ionosonde data spanning years 2014 and 2015, the authors examine the power, coherence and phase difference of perturbations in the daytime ionosphere and Pc5/Pi3 geomagnetic pulsations, distinguishing a subset of events during periods of magnetic quiescence and moderate magnetic storms with coherence greater than 0.5 from ULF wave signatures in the ionosphere observed under conditions favourable to strong geomagnetic storms. This extends previous studies by Pilipenko et al. (2014a and 2014b) that considered ULF wave-driven oscillations in the ionosphere F2 region during strong and small magnetic storms. Furthermore, the manuscript presents a new methodology to the automated detection of the foF2 critical frequency from ionograms that could be of interest for the research community working on determining factors that influence the amplitude and phase of perturbations in the ionosphere as these are detected on the ground. There are, however, several issues that hinder my recommendation of this manuscript for publication in Annales Geophysicae in its present form. There are major issues with the English language use, several typographical errors and in general, it is poorly written making it difficult to understand the scientific rationale behind this study.

Thank you very much for the comments. We are working on improvement of the language of the MS.

For example, in line 19, it reads: "Modulation of ionospheric parameters by Pc5 pulsations was reported : : :", without detailing which parameters are meant here. In the same line, it goes on to say: "Majority of publications are based on the radar observation : : :" (which would more correctly read "The majority of publications are based on radar observations : : :"), without making it clear to which publications the authors refer.

These points will be clarified

It would be worthwhile to establish in the Introduction the need for a study such as the present by listing past publications focused on perturbations in the ionosphere driven by ULF waves. Early results on geomagnetic pulsations in the

ULF wave frequency range associated with total electron content (TEC) fluctuations date back to 1976 and include the following: - Davies & Hartmann (1976), Short-period fluctuations in total columnar electron content, Journal of Geophysical Research, https://doi.org/10.1029/JA081i019p03431 - Okuzawa & Davies (1981), Pulsations in total columnar electron content, Journal of Geophysical Research, https://doi.org/10.1029/JA086iA03p01355

In the previous version, we have briefly mentioned only auroral Pc5 pulsations, while the papers by Davies (1976) and Okuzawa (1981) were devoted to Pc3-4 pulsations at lower latitudes. In the revision we plan to extend the Introduction section and include these and some other references.

Total electron content variations have been proven a powerful tool in the detection of ionospheric signatures of ULF waves at high latitudes as well as data from ionosondes exploiting the radio-wave reflecting properties of the ionosphere, as it is detailed by Watson et al. (2015). It is not clear to me and perhaps the reader how the results of Watson are different from those of Kozyreva et al. (2019) briefly mentioned in line 29. Nor the difference with those of Pilipenko et al. (2014b) derived from data collected during a different magnetic storm.

This analysis will be included into the Introduction section and to Discussion. We plan to give a more thorough analysis of observational results in the Introduction section and of physical mechanisms in the Discussion.

The following publications could be added to improve the placement of this work in the context of existing literature: - Baddeley et al. (2005), On the coupling between unstable magnetospheric particle populations and resonant high-m ULF wave signatures in the ionosphere, Annales Geophysicae, https://doi.org/10.5194/angeo-23-567-2005 - Buchert et al. (1999), Ionospheric conductivity modulation in ULF pulsations, Journal of Geophysical Research, https://doi.org/10.1029/1998JA900180

The references will be added to the MS

In lines 31 and 32, the authors note that the association of waves with moderate amplitudes with variations of the foF2 critical frequency have not been studied. However, how their amplitude is defined as moderate is not described nor later in the manuscript. As mentioned in the title of the manuscript, the reader is waiting for more details on these moderate geomagnetic pulsations, in my mind.

Thank you very much for this comment. In the next version the data analysis will be improved and a classification of the intervals in accordance with spectral power density at frequencies of PSD local maxima will be added. This will allow to quantify such terms as "moderate".

In lines 62 and 63, could the authors explain in quantitative terms how high the signal intensity at the reflection boundary should be as well as the amplitude ratio of the signal intensity at the reflection boundary to the power above it? Later, in lines 68 and 71, the authors note that a threshold for the time derivative of the foF2 critical frequency is calculated from the variance over a time interval of length t1. Is the variance of the foF2 critical frequency meant? How is the length of the time interval t1 defined?

The description of the approximation procedure will be extended. Besides, the parameters values used as the initial point of approximation will be added as a supplementary file.

Section 2.2 would benefit from an ionogram on which the described method has been used to detect the ionosphere F2 region critical frequency, clearly illustrating the new method for the foF2 critical frequency automated detection.

Figure 1, its capture, and the text explaining the procedure will be improved to make the detail of the approximation procedure clearer.

In Figures 4 and 7, it would be worthwhile to note the frequency of the primary and secondary maximum in power and provide further explanation at which frequency the coherence is taken for the statistics provided in Section 3.1.2.

The explanations will be added. Actually, in the present version, there is a difference in the examples, where 2 frequencies are used and statistical results where only the first spectral maxima are analyzed. This point will be improved in the revised version.

In Section 3.1.1, in addition to the details offered for the two intervals in March and July 2015, the two examples could be utilised to introduce the criteria set for selecting similar events for subsequent statistical analysis.

The classification of events will be improved. Really, in the present version, not identical criteria are used at different stages of data analysis. The choice of event class is not random but it may be difficult to discriminate between different types of events taken for comparison with coherent b-foF2 events in each case. The explicit classification will be given in the beginning of the Data processing section.

In Figures 9, 10, 12 and 13, as these are described in Section 3.1.2, what does "occurrence" and the symbol "D" mean in this context? Do the authors refer to "probability of occurrence"?

Yes, that is the empirical probability density, the term will be explicitly explained in the text

As they stand, the conclusions reached and briefly summarised in the first paragraph of Section 4 of this manuscript are a bit vague. Although it is suggested that this study is focused on variations of the ionosphere's critical frequency foF2 during quiet and moderately disturbed geomagnetic conditions, the most favourable values of the Dst index lay between -100 and -50 nT. Under such conditions, how often would it be expected to detect events are associated with ULF geomagnetic pulsations? How would the low occurrence rate (3%) of coherent events change if periods of highly disturbed conditions or quiescence were excluded? Please also consider commenting on the solar wind conditions that are favourable for the occurrence of coherent events and specifically, provide the range of solar wind speed and dynamic pressure values.

The new classification of all the intervals analyzed will give answers to all these questions. You are absolutely right, that in the previous version of our MS, the problems caused by the method of foF2 detection from the ionogram in the disturbed ionosphere can hardly be discriminated with the ionospheric Pc5/Pi3 occurrence probability. In the next version we shall limit ourselves with the disturbance levels, for which the detection procedure is valid and concentrate only on the intervals when quality of foF2 detection allows for the spectral analysis. For these intervals, we shall analyze the specific features of high coherent b-foF2 pulsations and space weather conditions favorable for their occurrence. Probabilities of coherent b-foF2 pulsations under favorable conditions will be given explicitly in the text.

Lastly, there are inconsistencies in the referencing style and specifically, on page 9 and 10, the year of publication in Mager et al. (2013), Min et al. (2017) and Viall et al. (2009) should be moved to the end of each reference.

The references will be corrected

Please also note the supplement to this comment:
https://angeo.copernicus.org/preprints/angeo-2020-16/angeo-2020-16-AC1-supplement.pdf

**Supplement:**

We thank both referees for the helpful comments. Point-by-point answers are given in blue.

**Anonymous Referee #1**

Received and published: 29 May 2020 Authors: N. Yagova et al.

Title: Even moderate geomagnetic pulsations can cause fluctuations of foF2 frequency of the auroral ionosphere.

The authors investigate the relation between colocated, simultaneous fluctuations in the F2 critical frequency and geomagnetic time series. They developed an automated method for inferring foF2 frequency from the ionograms. Events with foF2 frequencies modulated in the Pc5/Pi3 frequency range are analysed. The properties of a subset of events with coherence greater than 0.5 are compared to the average properties of the whole population of the events. The authors found that coherent events favour moderate geomagnetic conditions and show typical features of field line resonances. At the same time, it is noted that the automated detection of foF2 is not applicable to disturbed conditions. The paper, in general, is difficult to read and follow mainly because of its poor language. We strongly recommend the authors to use some spell-check tool to improve the quality of the presentation of their thoughts.

**Thank you very much for your comments. We plan to improve the language.**

A figure illustrating the automated

detection of the critical frequencies would be helpful for the reader.

We plan to extend the Figure 1 and the capture to explain the detection procedure in a more clear way.

**The focus of**

the paper is on coherent events, however, neither the 'event' is exactly defined, nor the significance level of the calculated coherence is given.

The significance levels will be added to the paper, and a more detailed classification of the events will be added. The more detailed classification will be added to clarify the basis for comparison of the coherent magnetic pulsations with the foF2 pulsations on each step of the data analysis.

There is also some inconsistency

in the paper about at what frequency the coherence is taken (f1 vs f2: statistics in Fig 10 vs. case studies ). Results presented in Fig 11 again suggest a link to f2 (at least based on the case studies).

This issue will be discussed in more details. While it is possible to analyze several frequencies in case studies, in statistics it can lead to an artificial enhancement of coherence between magnetic and foF2 pulsations. That is why, in the current version of the MS, we have used the comparison of only the first frequencies in statistics to obtain the lower boundary for the coherence estimates. In the next version, the data processing technique with both frequency maxima taken into account will be applied, and its influence on final statistical relationships will be considered.

The relative occurrence of coherent events is very low (\_ 3%). The statistics support that coherent events tend to occur under moderately disturbed geomagnetic and interplanetary conditions. However, the significance of this result is not clear due to 1. the low relative occurrence of coherent events, 2. the unknown significance level of coherence, 3. the limitations of foF2 detection under disturbed conditions, 4. the applied normalisation on which limited information is given. It was also not investigated how often the coherent events show up provided the conditions are favourable (moderate disturbance). Without this information the study is not complete and cannot be judged.

This summary of the problems in data analysis is really very important. We shall try to improve data analysis in accordance with the following plan.

- 1. A more detailed classification of the analyzed intervals will be applied, e.g. the intervals will be sorted into several sub-classes: in accordance with the 1) foF2 data availability ; 2) amplitudes of geomagnetic and foF2 pulsations; 3) coherence level between geomagnetic and foF2 pulsations
- 2. This will allow to estimate statistically the space weather effects for each group of intervals and to exclude the ambiguity which now exists in the analysis of highly disturbed intervals.
- 3. The number of analyzed events will be given for all the statistical studies and normalization procedure will be explained in more details.
- 4. Significance of the coherence estimate will be added

I recommend a major revision. Below I give a list of my minor comments in two groups. The first group relates to science, the second to the language. The latter is far from being complete. It would have been a long list. Minor comments on the scientific content

**We shall try to improve the text in the accordance with the minor remarks listed below. Specific remarks are given below to some questions which need more explanations.**

I 97: "about 10 nT and 0.08 MHz": Revise these values based on Fig 3!
I 100 what is the significance level for the coherence values in these calculations?
I 112: "about 80 nT and 0.08 MHz": although geomagnetic variations are several times greater here then for event 1, the foF2 variations are smaller. Comment?

**The problem of different efficiency of geomagnetic pulsations in foF2 modulation will be discussed. This might be explained by different spatial scales of pulsations.**

I 126: A MLT distribution of occurrence of the foF2 variations –> The MLT distribution of the occurrence of foF2 variations Under what conditions? What criteria define an event?

I 128: "frequency distributions of geomagnetic and foF2 pulsations": in general? I guess the distribution is based geomagnetic pulsation events simultaneous with foF2 events.

Figure 9: What is D (vertical axis)? Relative occurrence

Figure 10: Why the distribution of the first spectral peaks is presented. In your example events f2 has the higher coherence and corresponding Psw fluctuations. Are not your examples presented typical for the coupling between foF2 and geomagnetic variations?

In the present version, we have chosen this variant for statistical studies, because the analysis of different combinations of frequency maxima in foF2 and geomagnetic pulsations can lead to an overestimation of common features in their spectra. We understand, that our variant gives the underestimated level of

similarity. We have used this variant to obtain an estimate from the bottom for the similarity between the two types of pulsations. . In the next version, we will apply the data processing technique with both frequency maxima taken into account.

I 131: "spectral coherence at SOD" : at what frequency? coherence at f1?
I 132: Give the significance level!
Figure 10 b) Mark the significance level in this plot!
I 142: some information on the derivation of the weight functions and how they applied to normalize the data is needed
This information will be included into the text

I 153: "the 4-day minimum Dst and 6-hour maximal AE": intervals centred or preceding the coherent event?

All the parameters are given for the preceding intervals

I 158-162 Do coherent events occur under severely disturbed conditions, just they cannot be observed? Or they do not occur under those conditions at all? How does this observational limitation affect your conclusions?

This question will be answered using a more detailed classification of events (see our answer to the last point of the major comments). As for the ionosonde data, Pc5s in the F layer can be recorded under extremely disturbed conditions only rarely, because of blanketing or absorption below. This leads to the situation when case studies of rare Pc5 events may be possible, but the amount of data is not enough for statistical analysis.

I 165: "3-hour mean values of BZ and V and 3-hour maximal value of \_Psw": in which interval? (same issue as above)

Again, the preceding intervals are used

I 181: "Amplitude of SW dynamic pressure fluctuations show an association with occurrence of coherent foF2 \_ B pulsations": only 2 examples were presented in favour of this statement. Figure 13 b) does not yield any information on the spectral content of the pressure variations, and hence their relation to the coherent frequency. You seem to focus your statistics on f1 (first peak, e.g. Fig 10 a; coherence at f1), while your 2 examples had their relation with the SW pressure variations at f2.

The data of SW dynamical pressure have many gaps. That is why we only qualitatively consider some example events. In the future, we plan to study the cross-spectra of IMF and SW dynamic pressure fluctuations with foF2 pulsations based on an extended data set.

I 185: refer to your observations relevant presented in Fig 11. and show how they support the FLR nature of the coherent subset

We plan to add an example of FLR properties of coherent b-foF2 pulsations

I 193: "The picture changed dramatically": be more specific!

Thank you very much for the help with the text. We shall take all the comments into account.

Further comments:

I 1: "variations of the critical frequency": maybe "modulation" of the critical frequency could also be used here

I 1: o-mode radiowave -> o-mode radio waves

l 2: in 1–5â  $\,\check{}$  AL'mHz –> in the 1–5â  $\,\check{}$  AL'mHz

I 4: delete "daytime Pc5/Pi3 geomagnetic pulsations and" [foF2 is obviously not detected in geomagnetic pulsations]

I 6: at SOD station -> observed at SOD station

I 6: with the data of a station pair located at the same magnetic meridian -> using the data of a station pair located along the same magnetic meridian

I 8: Meanwhile, -> "At the same time," OR "However,"

I 8: "the analysis of geomagnetic and foF2 variations show intervals with noticeable coherence for both horizontal components" -> "the analysis of geomagnetic ad foF2 variations shows intervals of significant [OR remarkable] coherence with both horizontal geomagnetic components" [foF2 does not have any components]

| 11: averaged -> the average

l 11: coherent to -> coherent with

l 13: show -> shows

I 14: show -> shows [I suggest to use some synonym of 'show', such as 'reveal',

'indicate'. Use an online Thesaurus for finding synonyms]

l 19: Majority of publications are based on the radar observation -> Majority of publications on the topic are based on radar observations

I 20: of electron concentration at certain altitude -> of the electron concentration at a certain altitude

I 23: with mainly compressional mode of MHD wave in the magnetosphere -> with mainly compressional mode magnetospheric waves

I 26: An effect of TEC modulation by ULF wave -> The effect of TEC modulation by a ULF waves

l 27: and zones -> and also from zones

I 28: observed pulsations -> the observed pulsations [a large number of articles are missing from the text, check!]

I 31: the recovery phase of the magnetic storm —> the recovery phase of a magnetic storm

I 33: aimed on variations -> aimed at comparing variations

I 37: It makes an ionogram -> It obtains an ionogram recording

I 40: 10 s sampling rate -> 10 s sampling period/interval

I 40: and we also use the data of the MAS station, which is a part of IMAGE -> we also use data of recorded at MAS station of the IMAGE network

I 42: To analyzed -> To analyze

I 43: and also Dst and AE indexes are used -> as well as Dst and AE indexes

I 46: with quality and time resolution enough -> with good quality and time resolution is enough

I 56: for the reader's sake refer to your Fig 1 here.

I 58: Lorentsian -> Lorentzian

l 60: 235 km -> 235 km.

I 60: Coefficients f1, \_f = f2 \_ f1, k, and \_ are found as a result of fitting procedure,

described below. -> A fitting procedure described below is used to find f1,  $_f = f2 _ f1$ , k, and  $_. [f1,_f, _ are not coefficients]$  What are the meaning of f1 (I guess f at h1) and f2?

I 61: boundary is determined as a line -> boundary consists of a set of (h,f) points

I 62: Signal intensity I at the boundary should be high -> Signal intensity I is high

I 63: Amplitude ratio R of the signal intensity at the boundary line to the power above it should also be high -> The contrast between the peak and the background (characterized by the amplitude ratio R) is high [or similar, your version is confusing. Intensity to power ratio called amplitude ratio... It is not clear what is 'above'. At higher frequency?]
I 64: As four fitting factors are used -> We then fit Eq(1) to the detected boundary points. As four fitting factors are used

I 64: organized and a parameter -> organized. A parameter

I 65: over the "cross" in space of parameters -> over the parameter space [?]

I 65: where x is a point in the space of parameters, and i is a parameter number ->

where 'x' is a point in the parameter space, and 'i' identifies the parameter [and what is c? ]

I 64: Give a representative example, e.g. the values of the parameters used to derive the fits presented in Fig 1!

I 68: time dependence f(t): Do you mean the time dependence foF2(t)?

I 69: give a typical value of t1!

| 71: the other [??? or another]

I 73: Examples of approximation curves are given in Figure 1:

Figure 1: Complete the figure caption by including "the fitted curves are plotted over the ionograms in yellow" or similar. Add a reference to the fitting curves in Fig 1 in the main text, as well.

l 76: pictures -> plots

I 79: Note, that the ionograms are rotated by 90â °U e in respect to usual f \_ H presentation:

This sentence should come earlier! (with respect to)

I 80: foF2 -> foF2 values

I 84: Statistical analysis: Statistical analysis of what?

l 84: interval -> intervals

I 85: We studied -> We studied the effect/influence of .... on...

I 87: resolution, enough -> resolution high enough

I 90: Cross-spectra are calculated for foF2 variations, on one hand, and components of the geomagnetic field pulsations, on the other hand. -> Cross-spectra are calculated between foF2 variations and components of the geomagnetic field pulsations.

I 100: "at low frequency part of spectrum f < 2 mHz" -> "in the low frequency part (f < 2 mHz) of spectrum"

I 101: peak with maximal  $y^2 = 0.6 \rightarrow peak$  with  $y^2 = 0.6$

I 129: "with frequencies (f1 > 3.7 mHz)" -> "with frequencies above 3.7 mHz

I 130: "The distribution of Pc5/Pi3 intervals over foF2 \_ b spectral coherence at SOD are shown in Figure 10b for two" -> "The histogram of the foF2 \_ b spectral coherence at SOD is shown in Figure 10b for the two"

I 137: "a question arises about the pulsation properties and external parameters, favorable for their occurrence": rephrase!

I 138: "the geomagnetic pulsations" -> "a subset of the geomagnetic pulsations"

I 139: " with all the intervals, selected" -> " with all the events selected"

I 142: "calculated with the weight functions, which are found from" -> "calculated with weight functions derived from"

I 143: "coherent and pulsations and averaged" -> "coherent pulsations and averaged"

l 151: indexes -> indices

I 159: "limited by" -> "limited to"

l 186: "in coherent fo
F2 $\_$ Bx pulsations" : delete. This information is already given earlier in the sentence.

I 197: "For the first time, a statistical study of foF2 variations in Pc5/Pi3 range and their relation to geomagnetic pulsation in the conjugated position at SOD station and its spatial distribution along a magnetic meridian." Check the sentence (missing predicate).

**Anonymous Referee #2**

Received and published: 19 August 2020

In their manuscript "Even moderate geomagnetic pulsations can cause fluctuations of foF2 frequency of the auroral ionosphere", Yagova et al. explore variations of the ionosphere F2 region critical frequency (foF2) and ultra-low frequency (ULF) waves in the Pc5 and Pi3 frequency bands detected at auroral latitudes.

Using ground magnetometer and ionosonde data spanning years 2014 and 2015, the authors examine the power, coherence and phase difference of perturbations in the daytime ionosphere and Pc5/Pi3 geomagnetic pulsations, distinguishing a subset of events during periods of magnetic quiescence and moderate magnetic storms with coherence greater than 0.5 from ULF wave signatures in the ionosphere observed under conditions favourable to strong geomagnetic storms. This extends previous studies by Pilipenko et al. (2014a and 2014b) that considered ULF wave-driven oscillations in the ionosphere F2 region during strong and small magnetic storms.

Furthermore, the manuscript presents a new methodology to the automated detection of the foF2 critical frequency from ionograms that could be of interest for the research community working on determining factors that influence the amplitude and phase of perturbations in the ionosphere as these are detected on the ground. There are, however, several issues that hinder my recommendation of this manuscript for publication in Annales Geophysicae in its present form.

There are major issues with the English language use, several typographical errors and in general, it is poorly written making it difficult to understand the scientific rationale behind this study.

Thank you very much for the comments. We are working on improvement of the language of the MS.

For example, in line 19, it reads: "Modulation of ionospheric parameters by Pc5 pulsations was reported : : :", without detailing which parameters are meant here. In the same line, it goes on to say: "Majority of publications are based on the radar observation : : :" (which would more correctly read "The majority of publications are based on radar observations : : :"), without making it clear to which publications the authors refer.

**These points will be clarified**

It would be worthwhile to establish in the Introduction the need for a study such as the present by listing past publications focused on perturbations in the ionosphere driven by ULF waves. Early results on geomagnetic pulsations in the ULF wave frequency range associated with total electron content (TEC) fluctuations date back to 1976 and include the following:

Davies & Hartmann (1976), Short-period fluctuations in total columnar electron content, Journal of Geophysical Research, https://doi.org/10.1029/JA081i019p03431
Okuzawa & Davies (1981), Pulsations in total columnar electron content, Journal of Geophysical Research, https://doi.org/10.1029/JA086iA03p01355

**In the previous version, we have briefly mentioned only auroral Pc5 pulsations, while the papers by Davies (1976) and Okuzawa (1981) were devoted to Pc3-4 pulsations at lower latitudes. In the revision we plan to extend the Introduction section and include these and some other references.**

Total electron content variations have been proven a powerful tool in the detection of ionospheric signatures of ULF waves at high latitudes as well as data from ionosondes exploiting the radio-wave reflecting properties of the ionosphere, as it is detailed by Watson et al. (2015). It is not clear to me and perhaps the reader how the results of Watson are different from those of Kozyreva et al. (2019) briefly mentioned in line 29. Nor the difference with those of Pilipenko et al. (2014b) derived from data collected during a different magnetic storm.

**This analysis will be included into the Introduction section and to Discussion. We plan to give a more thorough analysis of observational results in the Introduction section and of physical mechanisms in the Discussion.**

The following publications could be added to improve the placement of this work in the context of existing literature:

Baddeley et al. (2005), On the coupling between unstable magnetospheric particle populations and resonant high-m ULF wave signatures in the ionosphere, Annales Geophysicae, https://doi.org/10.5194/angeo-23-567-2005
 Buchert et al. (1999), Ionospheric conductivity modulation in ULF pulsations, Journal

of Geophysical Research, https://doi.org/10.1029/1998JA900180

**The references will be added to the MS**

In lines 31 and 32, the authors note that the association of waves with moderate amplitudes with variations of the foF2 critical frequency have not been studied. However, how their amplitude is defined as moderate is not described nor later in the manuscript. As mentioned in the title of the manuscript, the reader is waiting for more details on these moderate geomagnetic pulsations, in my mind.

**Thank you very much for this comment. In the next version the data analysis will be improved and a classification of the intervals in accordance with spectral power density at frequencies of PSD local maxima will be added. This will allow to quantify such terms as "moderate".**

In lines 62 and 63, could the authors explain in quantitative terms how high the signal intensity at the reflection boundary should be as well as the amplitude ratio of the signal intensity at the reflection boundary to the power above it? Later, in lines 68 and 71, the authors note that a threshold for the time derivative of the foF2 critical frequency is calculated from the variance over a time interval of length t1. Is the variance of the foF2 critical frequency meant? How is the length of the time interval t1 defined? The description of the approximation procedure will be extended. Besides, the parameters values used as the initial point of approximation will be added as a supplementary file.

Section 2.2 would benefit from an ionogram on which the described method has been used to detect the ionosphere F2 region critical frequency, clearly illustrating the new method for the foF2 critical frequency automated detection.

Figure 1, its capture, and the text explaining the procedure will be improved to make the detail of the approximation procedure clearer.

In Figures 4 and 7, it would be worthwhile to note the frequency of the primary and secondary maximum in power and provide further explanation at which frequency the coherence is taken for the statistics provided in Section 3.1.2.

The explanations will be added. Actually, in the present version, there is a difference in the examples, where 2 frequencies are used and statistical results where only the first spectral maxima are analyzed. This point will be improved in the revised version.

In Section 3.1.1, in addition to the details offered for the two intervals in March and July 2015, the two examples could be utilised to introduce the criteria set for selecting similar events for subsequent statistical analysis.

The classification of events will be improved. Really, in the present version, not identical criteria are used at different stages of data analysis. The choice of event class is not random but it may be difficult to discriminate between different types of events taken for comparison with coherent b-foF2 events in each case. The explicit classification will be given in the beginning of the Data processing section.

In Figures 9, 10, 12 and 13, as these are described in Section 3.1.2, what does "occurrence" and the symbol "D" mean in this context? Do the authors refer to "probability of occurrence"?

**Yes, that is the empirical probability density, the term will be explicitly explained in the text**

As they stand, the conclusions reached and briefly summarised in the first paragraph of Section 4 of this manuscript are a bit vague. Although it is suggested that this study is focused on variations of the ionosphere's critical frequency foF2 during quiet and moderately disturbed geomagnetic conditions, the most favourable values of the Dst index lay between -100 and -50 nT. Under such conditions, how often would it be expected to detect events are associated with ULF geomagnetic pulsations? How would the low occurrence rate (3%) of coherent events change if periods of highly disturbed conditions or quiescence were excluded? Please also consider commenting on the solar wind conditions that are favourable for the occurrence of coherent events and specifically, provide the range of solar wind speed and dynamic pressure values.

The new classification of all the intervals analyzed will give answers to all these questions. You are absolutely right, that in the previous version of our MS, the problems caused by the method of foF2 detection from the ionogram in the disturbed ionosphere can hardly be discriminated with the ionospheric Pc5/Pi3 occurrence probability. In the next version we shall limit ourselves with the disturbance levels, for which the detection procedure is valid and concentrate only on the intervals when quality of foF2 detection

allows for the spectral analysis. For these intervals, we shall analyze the specific features of high coherent bfoF2 pulsations and space weather conditions favorable for their occurrence. Probabilities of coherent bfoF2 pulsations under favorable conditions will be given explicitly in the text.

Lastly, there are inconsistencies in the referencing style and specifically, on page 9 and 10, the year of publication in Mager et al. (2013), Min et al. (2017) and Viall et al. (2009) should be moved to the end of each reference.

The references will be corrected

---

## Author Comment (AC2) · 14 Sep 2020

Dear Referee, thank you very much for your helpful report. Point-by-point answers are given below.

We thank both referees for the helpful comments. Point-by-point answers are given in blue.

Anonymous Referee #1 Authors: N. Yagova et al. Title: Even moderate geomagnetic pulsations can cause fluctuations of foF2 frequency of the auroral ionosphere. The authors investigate the relation between

colocated, simultaneous fluctuations in the F2 critical frequency and geomagnetic time series. They developed an automated method for inferring foF2 frequency from the ionograms. Events with foF2 frequencies modulated in the Pc5/Pi3 frequency range are analysed. The properties of a subset of events with coherence greater than 0.5 are compared to the average properties of the whole population of the events. The authors found that coherent events favour moderate geomagnetic conditions and show typical features of field line resonances. At the same time, it is noted that the automated detection of foF2 is not applicable to disturbed conditions. The paper, in general, is difficult to read and follow mainly because of its poor language. We strongly recommend the authors to use some spell-check tool to improve the quality of the presentation of their thoughts.

Thank you very much for your comments. We plan to improve the language. A figure illustrating the automated detection of the critical frequencies would be helpful for the reader.

We plan to extend the Figure 1 and the capture to explain the detection procedure in a more clear way.

The focus of the paper is on coherent events, however, neither the 'event' is exactly defined, nor the significance level of the calculated coherence is given.

The significance levels will be added to the paper, and a more detailed classification of the events will be added. The more detailed classification will be added to clarify the basis for comparison of the coherent magnetic pulsations with the foF2 pulsations on each step of the data analysis.

There is also some inconsistency in the paper about at what frequency the coherence is taken (f1 vs f2: statistics in Fig 10 vs. case studies ). Results presented in Fig 11 again suggest a link to f2 (at least based on the case studies).

This issue will be discussed in more details. While it is possible to analyze several

frequencies in case studies, in statistics it can lead to an artificial enhancement of coherence between magnetic and foF2 pulsations. That is why, in the current version of the MS, we have used the comparison of only the first frequencies in statistics to obtain the lower boundary for the coherence estimates. In the next version, the data processing technique with both frequency maxima taken into account will be applied, and its influence on final statistical relationships will be considered.

The relative occurrence of coherent events is very low (_ 3%). The statistics support that coherent events tend to occur under moderately disturbed geomagnetic and interplanetary conditions. However, the significance of this result is not clear due to 1. the low relative occurrence of coherent events, 2. the unknown significance level of coherence, 3. the limitations of foF2 detection under disturbed conditions, 4. the applied normalisation on which limited information is given. It was also not investigated how often the coherent events show up provided the conditions are favourable (moderate disturbance). Without this information the study is not complete and cannot be judged.

This summary of the problems in data analysis is really very important. We shall try to improve data analysis in accordance with the following plan. 1. A more detailed classification of the analyzed intervals will be applied, e.g. the intervals will be sorted into several sub-classes: in accordance with the 1) foF2 data availability ; 2) amplitudes of geomagnetic and foF2 pulsations; 3) coherence level between geomagnetic and foF2 pulsations

2. This will allow to estimate statistically the space weather effects for each group of intervals and to exclude the ambiguity which now exists in the analysis of highly disturbed intervals.

3. The number of analyzed events will be given for all the statistical studies and normalization procedure will be explained in more details. 4. Significance of the coherence estimate will be added

I recommend a major revision. Below I give a list of my minor comments in two groups.

The first group relates to science, the second to the language. The latter is far from being complete. It would have been a long list. Minor comments on the scientific content

We shall try to improve the text in the accordance with the minor remarks listed below. Specific remarks are given below to some questions which need more explanations. l 97: "about 10 nT and 0.08 MHz": Revise these values based on Fig 3! l 100 what is the significance level for the coherence values in these calculations? l 112: "about 80 nT and 0.08 MHz": although geomagnetic variations are several times greater here then for event 1, the foF2 variations are smaller. Comment?

The problem of different efficiency of geomagnetic pulsations in foF2 modulation will be discussed. This might be explained by different spatial scales of pulsations.

l 126: A MLT distribution of occurrence of the foF2 variations –> The MLT distribution of the occurrence of foF2 variations Under what conditions? What criteria define an event? l 128: "frequency distributions of geomagnetic and foF2 pulsations": in general? I guess the distribution is based geomagnetic pulsation events simultaneous with foF2 events.

Figure 9: What is D (vertical axis)? Relative occurrence

Figure 10: Why the distribution of the first spectral peaks is presented. In your example events f2 has the higher coherence and corresponding Psw fluctuations. Are not your examples presented typical for the coupling between foF2 and geomagnetic variations?

In the present version, we have chosen this variant for statistical studies, because the analysis of different combinations of frequency maxima in foF2 and geomagnetic pulsations can lead to an overestimation of common features in their spectra. We understand, that our variant gives the underestimated level of similarity. We have used this variant to obtain an estimate from the bottom for the similarity between the two types of pulsations. . In the next version, we will apply the data processing technique

with both frequency maxima taken into account.

l 131: "spectral coherence at SOD" : at what frequency? coherence at f1? l 132: Give the significance level! Figure 10 b) Mark the significance level in this plot! l 142: some information on the derivation of the weight functions and how they applied to normalize the data is needed This information will be included into the text

l 153: "the 4-day minimum Dst and 6-hour maximal AE": intervals centred or preceding the coherent event?

All the parameters are given for the preceding intervals

l 158-162 Do coherent events occur under severely disturbed conditions, just they cannot be observed? Or they do not occur under those conditions at all? How does this observational limitation affect your conclusions?

This question will be answered using a more detailed classification of events (see our answer to the last point of the major comments). As for the ionosonde data, Pc5s in the F layer can be recorded under extremely disturbed conditions only rarely, because of blanketing or absorption below. This leads to the situation when case studies of rare Pc5 events may be possible, but the amount of data is not enough for statistical analysis.

l 165: "3-hour mean values of BZ and V and 3-hour maximal value of _Psw": in which interval? (same issue as above)

Again, the preceding intervals are used

l 181: "Amplitude of SW dynamic pressure fluctuations show an association with occurrence of coherent foF2 _ B pulsations": only 2 examples were presented in favour of this statement. Figure 13 b) does not yield any information on the spectral content of the pressure variations, and hence their relation to the coherent frequency. You seem to focus your statistics on f1 (first peak, e.g. Fig 10 a; coherence at f1), while your 2 examples had their relation with the SW pressure variations at f2.

The data of SW dynamical pressure have many gaps. That is why we only qualitatively consider some example events. In the future, we plan to study the cross-spectra of IMF and SW dynamic pressure fluctuations with foF2 pulsations based on an extended data set.

l 185: refer to your observations relevant presented in Fig 11. and show how they support the FLR nature of the coherent subset

We plan to add an example of FLR properties of coherent b-foF2 pulsations

l 193: "The picture changed dramatically": be more specific!

Thank you very much for the help with the text. We shall take all the comments into account. Further comments: l 1: "variations of the critical frequency": maybe "modulation" of the critical frequency could also be used here l 1: o-mode radiowave –> o-mode radio waves l 2: in 1–5â ËŸ AĽmHz –> in the 1–5â ËŸ AĽmHz l 4: delete "daytime Pc5/Pi3 geomagnetic pulsations and" [foF2 is obviously not detected in geomagnetic pulsations] l 6: at SOD station –> observed at SOD station l 6: with the data of a station pair located at the same magnetic meridian –> using the data of a station pair located along the same magnetic meridian l 8: Meanwhile, –> "At the same time," OR "However," l 8: "the analysis of geomagnetic and foF2 variations show intervals with noticeable coherence for both horizontal components" –> "the analysis of geomagnetic ad foF2 variations shows intervals of significant [OR remarkable] coherence with both horizontal geomagnetic components" [foF2 does not have any components] l 11: averaged –> the average l 11: coherent to –> coherent with l 13: show –> shows l 14: show –> shows [I suggest to use some synonym of 'show', such as 'reveal', 'indicate'. Use an online Thesaurus for finding synonyms] l 19: Majority of publications are based on the radar observation –> Majority of publications on the topic are based on radar observations l 20: of electron concentration at certain altitude –> of the electron concentration at a certain altitude l 23: with mainly compressional mode of MHD wave in the magnetosphere –> with mainly compressional mode

magnetospheric waves I 26: An effect of TEC modulation by ULF wave –> The effect of TEC modulation by a ULF waves I 27: and zones –> and also from zones I 28: observed pulsations –> the observed pulsations [a large number of articles are missing from the text, check!] I 31: the recovery phase of the magnetic storm –> the recovery phase of a magnetic storm I 33: aimed on variations –> aimed at comparing variations I 37: It makes an ionogram –> It obtains an ionogram recording I 40: 10 s sampling rate –> 10 s sampling period/interval I 40: and we also use the data of the MAS station, which is a part of IMAGE –> we also use data of recorded at MAS station of the IMAGE network I 42: To analyzed –> To analyze I 43: and also Dst and AE indexes are used –> as well as Dst and AE indexes I 46: with quality and time resolution enough –> with good quality and time resolution is enough I 56: for the reader's sake refer to your Fig 1 here. I 58: Lorentsian –> Lorentzian I 60: 235 km –> 235 km. I 60: Coefficients f1, _f = f2 _ f1, k, and _ are found as a result of fitting procedure, described below. –> A fitting procedure described below is used to find f1, _f = f2 _ f1, k, and _. [f1,_f, _ are not coefficients] What are the meaning of f1 (I guess f at h1) and f2? I 61: boundary is determined as a line –> boundary consists of a set of (h,f) points I 62: Signal intensity I at the boundary should be high –> Signal intensity I is high I 63: Amplitude ratio R of the signal intensity at the boundary line to the power above it should also be high –> The contrast between the peak and the background (characterized by the amplitude ratio R) is high [or similar, your version is confusing. Intensity to power ratio called amplitude ratio... It is not clear what is 'above'. At higher frequency?] I 64: As four fitting factors are used –> We then fit Eq(1) to the detected boundary points. As four fitting factors are used I 64: organized and a parameter –> organized. A parameter I 65: over the "cross" in space of parameters –> over the parameter space [?] I 65: where x is a point in the space of parameters, and i is a parameter number –> where 'x' is a point in the parameter space, and 'i' identifies the parameter [and what is c? ] I 64: Give a representative example, e.g. the values of the parameters used to derive the fits presented in Fig 1! I 68: time dependence f(t): Do you mean the time dependence foF2(t)? I 69: give a typical value of t1! I 71:

the other [??? or another] l 73: Examples of approximation curves are given in Figure 1: Figure 1: Complete the figure caption by including "the fitted curves are plotted over the ionograms in yellow" or similar. Add a reference to the fitting curves in Fig 1 in the main text, as well. l 76: pictures –> plots l 79: Note, that the ionograms are rotated by 90â °U ËŻe in respect to usual f _ H presentation: This sentence should come earlier! (with respect to) l 80: foF2 –> foF2 values l 84: Statistical analysis: Statistical analysis of what? l 84: interval –> intervals l 85: We studied –> We studied the effect/influence of .... on... l 87: resolution, enough –> resolution high enough l 90: Cross-spectra are calculated for foF2 variations, on one hand, and components of the geomagnetic field pulsations, on the other hand. –> Cross-spectra are calculated between foF2 variations and components of the geomagnetic field pulsations. l 100: "at low frequency part of spectrum f < 2 mHz" –> "in the low frequency part (f < 2 mHz) of spectrum" l 101: peak with maximal y2 = 0.6 –> peak with y2 = 0.6 l 129: "with frequencies (f1 > 3.7 mHz)" –> "with frequencies above 3.7 mHz l 130: "The distribution of Pc5/Pi3 intervals over foF2 _ b spectral coherence at SOD are shown in Figure 10b for two" –> "The histogram of the foF2 _ b spectral coherence at SOD is shown in Figure 10b for the two" l 137: "a question arises about the pulsation properties and external parameters, favorable for their occurrence": rephrase! l 138: "the geomagnetic pulsations" –> "a subset of the geomagnetic pulsations" l 139: " with all the intervals, selected" –> " with all the events selected" l 142: "calculated with the weight functions, which are found from" –> "calculated with weight functions derived from" l 143: "coherent and pulsations and averaged" –> "coherent pulsations and averaged" l 151: indexes –> indices l 159: "limited by" –> "limited to" l 186: "in coherent foF2 _ Bx pulsations" : delete. This information is already given earlier in the sentence. l 197: "For the first time, a statistical study of foF2 variations in Pc5/Pi3 range and their relation to geomagnetic pulsation in the conjugated position at SOD station and its spatial distribution along a magnetic meridian." Check the sentence (missing predicate).

Please also note the supplement to this comment:
https://angeo.copernicus.org/preprints/angeo-2020-16/angeo-2020-16-AC2-supplement.pdf

**Supplement:**

We thank both referees for the helpful comments. Point-by-point answers are given in blue.

**Anonymous Referee #1**

Authors: N. Yagova et al.
Title: Even moderate geomagnetic pulsations can cause fluctuations of foF2 frequency of the auroral ionosphere.
The authors investigate the relation between colocated, simultaneous fluctuations in the F2 critical frequency and geomagnetic time series. They developed an automated method for inferring foF2 frequency from the ionograms. Events with foF2 frequencies modulated in the Pc5/Pi3 frequency range are analysed. The properties of a subset of events with coherence greater than 0.5 are compared to the average properties of the whole population of the events. The authors found that coherent events favour moderate geomagnetic conditions and show typical features of field line resonances. At the same time, it is noted that the automated detection of foF2 is not applicable to disturbed conditions. The paper, in general, is difficult to read and follow mainly because of its poor language. We strongly recommend the authors to use some spell-check tool to improve the quality of the presentation of their thoughts.

Thank you very much for your comments. We plan to improve the language.
A figure illustrating the automated
detection of the critical frequencies would be helpful for the reader.

We plan to extend the Figure 1 and the capture to explain the detection procedure in a more clear way.

 The focus of
the paper is on coherent events, however, neither the 'event' is exactly defined, nor the
significance level of the calculated coherence is given.

The significance levels will be added to the paper, and a more detailed classification of the events will be added. The more detailed classification will be added to clarify the basis for comparison of the coherent magnetic pulsations with the foF2 pulsations on each step of the data analysis.

There is also some inconsistency
in the paper about at what frequency the coherence is taken (f1 vs f2: statistics
in Fig 10 vs. case studies ). Results presented in Fig 11 again suggest a link to f2 (at
least based on the case studies).

This issue will be discussed in more details. While it is possible to analyze several frequencies in case studies, in statistics it can lead to an artificial enhancement of coherence between magnetic and foF2 pulsations. That is why, in the current version of the MS, we have used the comparison of only the first frequencies in statistics to obtain the lower boundary for the coherence estimates. In the next version, the data processing technique with both frequency maxima taken into account will be applied, and its influence on final statistical relationships will be considered.

The relative occurrence of coherent events is very
low (_ 3%). The statistics support that coherent events tend to occur under moderately
disturbed geomagnetic and interplanetary conditions. However, the significance
of this result is not clear due to 1. the low relative occurrence of coherent events, 2.

the unknown significance level of coherence, 3. the limitations of foF2 detection under disturbed conditions, 4. the applied normalisation on which limited information is given. It was also not investigated how often the coherent events show up provided the conditions are favourable (moderate disturbance). Without this information the study is not complete and cannot be judged.

This summary of the problems in data analysis is really very important. We shall try to improve data analysis in accordance with the following plan.
1.  A more detailed classification of the analyzed intervals will be applied, e.g. the intervals will be sorted into several sub-classes: in accordance with the 1) foF2 data availability ; 2) amplitudes of geomagnetic and foF2 pulsations; 3) coherence level between geomagnetic and foF2 pulsations

2.  This will allow to estimate statistically the space weather effects for each group of intervals and to exclude the ambiguity which now exists in the analysis of highly disturbed intervals.

3.  The number of analyzed events will be given for all the statistical studies and normalization procedure will be explained in more details.
4.  Significance of the coherence estimate will be added

I recommend a major revision. Below I give a list
of my minor comments in two groups. The first group relates to science, the second to
the language. The latter is far from being complete. It would have been a long list.
Minor comments on the scientific content

We shall try to improve the text in the accordance with the minor remarks listed below. Specific remarks are given below to some questions which need more explanations.
l 97: "about 10 nT and 0.08 MHz": Revise these values based on Fig 3!
l 100 what is the significance level for the coherence values in these calculations?
l 112: "about 80 nT and 0.08 MHz": although geomagnetic variations are several times greater here then for event 1, the foF2 variations are smaller. Comment?

The problem of different efficiency of geomagnetic pulsations in foF2 modulation will be discussed. This might be explained by different spatial scales of pulsations.

l 126: A MLT distribution of occurrence of the foF2 variations –> The MLT distribution of the occurrence of foF2 variations Under what conditions? What criteria define an event?
l 128: "frequency distributions of geomagnetic and foF2 pulsations": in general? I guess the distribution is based geomagnetic pulsation events simultaneous with foF2 events.

Figure 9: What is D (vertical axis)? Relative occurrence

Figure 10: Why the distribution of the first spectral peaks is presented. In your example events f2 has the higher coherence and corresponding Psw fluctuations. Are not your examples presented typical for the coupling between foF2 and geomagnetic variations?

In the present version, we have chosen this variant for statistical studies, because the analysis of different combinations of frequency maxima in foF2 and geomagnetic pulsations can lead to an overestimation of common features in their spectra. We understand, that our variant gives the underestimated level of

similarity. We have used this variant to obtain an estimate from the bottom for the similarity between the two types of pulsations. . In the next version, we will apply the data processing technique with both frequency maxima taken into account.

l 131: "spectral coherence at SOD" : at what frequency? coherence at f1?
l 132: Give the significance level!
Figure 10 b) Mark the significance level in this plot!
l 142: some information on the derivation of the weight functions and how they applied to normalize the data is needed

This information will be included into the text

l 153: "the 4-day minimum Dst and 6-hour maximal AE": intervals centred or preceding the coherent event?

All the parameters are given for the preceding intervals

l 158-162 Do coherent events occur under severely disturbed conditions, just they cannot be observed? Or they do not occur under those conditions at all? How does this observational limitation affect your conclusions?

This question will be answered using a more detailed classification of events (see our answer to the last point of the major comments). As for the ionosonde data, Pc5s in the F layer can be recorded under extremely disturbed conditions only rarely, because of blanketing or absorption below. This leads to the situation when case studies of rare Pc5 events may be possible, but the amount of data is not enough for statistical analysis.

l 165: "3-hour mean values of BZ and V and 3-hour maximal value of _Psw": in which interval? (same issue as above)

Again, the preceding intervals are used

l 181: "Amplitude of SW dynamic pressure fluctuations show an association with occurrence of coherent foF2 _ B pulsations": only 2 examples were presented in favour of this statement. Figure 13 b) does not yield any information on the spectral content of the pressure variations, and hence their relation to the coherent frequency. You seem to focus your statistics on f1 (first peak, e.g. Fig 10 a; coherence at f1), while your 2 examples had their relation with the SW pressure variations at f2.

The data of SW dynamical pressure have many gaps. That is why we only qualitatively consider some example events. In the future, we plan to study the cross-spectra of IMF and SW dynamic pressure fluctuations with foF2 pulsations based on an extended data set.

l 185: refer to your observations relevant presented in Fig 11. and show how they support the FLR nature of the coherent subset

We plan to add an example of FLR properties of coherent b-foF2 pulsations

l 193: "The picture changed dramatically": be more specific!

Thank you very much for the help with the text. We shall take all the comments into account.
Further comments:
l 1: "variations of the critical frequency": maybe "modulation" of the critical frequency could also be used here
l 1: o-mode radiowave –> o-mode radio waves
l 2: in 1–5â ˘ AL'mHz –> in the 1–5â ˘ AL'mHz
l 4: delete "daytime Pc5/Pi3 geomagnetic pulsations and" [foF2 is obviously not detected in geomagnetic pulsations]
l 6: at SOD station –> observed at SOD station
l 6: with the data of a station pair located at the same magnetic meridian –> using the data of a station pair located along the same magnetic meridian
l 8: Meanwhile, –> "At the same time," OR "However,"
l 8: "the analysis of geomagnetic and foF2 variations show intervals with noticeable coherence for both horizontal components" –> "the analysis of geomagnetic ad foF2 variations shows intervals of significant [OR remarkable] coherence with both horizontal geomagnetic components" [foF2 does not have any components]
l 11: averaged –> the average
l 11: coherent to –> coherent with
l 13: show –> shows
l 14: show –> shows [I suggest to use some synonym of 'show', such as 'reveal', 'indicate'. Use an online Thesaurus for finding synonyms]
l 19: Majority of publications are based on the radar observation –> Majority of publications on the topic are based on radar observations
l 20: of electron concentration at certain altitude –> of the electron concentration at a certain altitude
l 23: with mainly compressional mode of MHD wave in the magnetosphere –> with mainly compressional mode magnetospheric waves
l 26: An effect of TEC modulation by ULF wave –> The effect of TEC modulation by a ULF waves
l 27: and zones –> and also from zones
l 28: observed pulsations –> the observed pulsations [a large number of articles are missing from the text, check!]
l 31: the recovery phase of the magnetic storm –> the recovery phase of a magnetic storm
l 33: aimed on variations –> aimed at comparing variations
l 37: It makes an ionogram –> It obtains an ionogram recording
l 40: 10 s sampling rate –> 10 s sampling period/interval
l 40: and we also use the data of the MAS station, which is a part of IMAGE –> we also use data of recorded at MAS station of the IMAGE network
l 42: To analyzed –> To analyze
l 43: and also Dst and AE indexes are used –> as well as Dst and AE indexes
l 46: with quality and time resolution enough –> with good quality and time resolution is enough
l 56: for the reader's sake refer to your Fig 1 here.
l 58: Lorentsian –> Lorentzian
l 60: 235 km –> 235 km.
l 60: Coefficients f1, _f = f2 _ f1, k, and _ are found as a result of fitting procedure,

described below. –> A fitting procedure described below is used to find f1, _f = f2 _ f1, k, and _. [f1,_f, _ are not coefficients] What are the meaning of f1 (I guess f at h1) and f2?

l 61: boundary is determined as a line –> boundary consists of a set of (h,f) points

l 62: Signal intensity I at the boundary should be high –> Signal intensity I is high

l 63: Amplitude ratio R of the signal intensity at the boundary line to the power above it should also be high –> The contrast between the peak and the background (characterized by the amplitude ratio R) is high [or similar, your version is confusing. Intensity to power ratio called amplitude ratio... It is not clear what is 'above'. At higher frequency?]

l 64: As four fitting factors are used –> We then fit Eq(1) to the detected boundary points. As four fitting factors are used

l 64: organized and a parameter –> organized. A parameter

l 65: over the "cross" in space of parameters –> over the parameter space [?]

l 65: where x is a point in the space of parameters, and i is a parameter number –> where 'x' is a point in the parameter space, and 'i' identifies the parameter [and what is c? ]

l 64: Give a representative example, e.g. the values of the parameters used to derive the fits presented in Fig 1!

l 68: time dependence f(t): Do you mean the time dependence foF2(t)?

l 69: give a typical value of t1!

l 71: the other [??? or another]

l 73: Examples of approximation curves are given in Figure 1:

Figure 1: Complete the figure caption by including "the fitted curves are plotted over the ionograms in yellow" or similar. Add a reference to the fitting curves in Fig 1 in the main text, as well.

l 76: pictures –> plots

l 79: Note, that the ionograms are rotated by 90â °U ¸e in respect to usual f _ H presentation: This sentence should come earlier! (with respect to)

l 80: foF2 –> foF2 values

l 84: Statistical analysis: Statistical analysis of what?

l 84: interval –> intervals

l 85: We studied –> We studied the effect/influence of .... on...

l 87: resolution, enough –> resolution high enough

l 90: Cross-spectra are calculated for foF2 variations, on one hand, and components of the geomagnetic field pulsations, on the other hand. –> Cross-spectra are calculated between foF2 variations and components of the geomagnetic field pulsations.

l 100: "at low frequency part of spectrum f < 2 mHz" –> "in the low frequency part (f < 2 mHz) of spectrum"

l 101: peak with maximal y2 = 0.6 –> peak with y2 = 0.6

l 129: "with frequencies (f1 > 3.7 mHz)" –> "with frequencies above 3.7 mHz

l 130: "The distribution of Pc5/Pi3 intervals over foF2 _ b spectral coherence at SOD are shown in Figure 10b for two" –> "The histogram of the foF2 _ b spectral coherence at SOD is shown in Figure 10b for the two"

l 137: "a question arises about the pulsation properties and external parameters, favorable for their occurrence": rephrase!

l 138: "the geomagnetic pulsations" –> "a subset of the geomagnetic pulsations"

l 139: " with all the intervals, selected" –> " with all the events selected"

l 142: "calculated with the weight functions, which are found from" –> "calculated with weight functions derived from"

l 143: "coherent and pulsations and averaged" –> "coherent pulsations and averaged"

l 151: indexes –> indices
l 159: "limited by" –> "limited to"
l 186: "in coherent foF2 _ Bx pulsations" : delete. This information is already given earlier in the sentence.
l 197: "For the first time, a statistical study of foF2 variations in Pc5/Pi3 range and their relation to geomagnetic pulsation in the conjugated position at SOD station and its spatial distribution along a magnetic meridian." Check the sentence (missing predicate).

**Anonymous Referee #2**

In their manuscript "Even moderate geomagnetic pulsations can cause fluctuations of foF2 frequency of the auroral ionosphere", Yagova et al. explore variations of the ionosphere F2 region critical frequency (foF2) and ultra-low frequency (ULF) waves in the Pc5 and Pi3 frequency bands detected at auroral latitudes.
Using ground magnetometer and ionosonde data spanning years 2014 and 2015, the authors examine the power, coherence and phase difference of perturbations in the daytime ionosphere and Pc5/Pi3 geomagnetic pulsations, distinguishing a subset of events during periods of magnetic quiescence and moderate magnetic storms with coherence greater than 0.5 from ULF wave signatures in the ionosphere observed under conditions favourable to strong geomagnetic storms. This extends previous studies by Pilipenko et al. (2014a and 2014b) that considered ULF wave-driven oscillations in the ionosphere F2 region during strong and small magnetic storms.
Furthermore, the manuscript presents a new methodology to the automated detection of the foF2 critical frequency from ionograms that could be of interest for the research community working on determining factors that influence the amplitude and phase of perturbations in the ionosphere as these are detected on the ground. There are, however, several issues that hinder my recommendation of this manuscript for publication in Annales Geophysicae in its present form.
There are major issues with the English language use, several typographical errors and in general, it is poorly written making it difficult to understand the scientific rationale behind this study.

Thank you very much for the comments. We are working on improvement of the language of the MS.

For example, in line 19, it reads: "Modulation of ionospheric parameters by Pc5 pulsations was reported : : :", without detailing which parameters are meant here. In the same line, it goes on to say: "Majority of publications are based on the radar observation : : :" (which would more correctly read "The majority of publications are based on radar observations : : :"), without making it clear to which publications the authors refer.

These points will be clarified

It would be worthwhile to establish in the Introduction the need for a study such as the present by listing past publications focused on perturbations in the ionosphere driven by ULF waves. Early results on geomagnetic pulsations in the ULF wave frequency range associated with total electron content (TEC) fluctuations date back to 1976 and include the following:

- Davies & Hartmann (1976), Short-period fluctuations in total columnar electron content, Journal of Geophysical Research, https://doi.org/10.1029/JA081i019p03431
- Okuzawa & Davies (1981), Pulsations in total columnar electron content, Journal of Geophysical Research, https://doi.org/10.1029/JA086iA03p01355

*In the previous version, we have briefly mentioned only auroral Pc5 pulsations, while the papers by Davies (1976) and Okuzawa (1981) were devoted to Pc3-4 pulsations at lower latitudes. In the revision we plan to extend the Introduction section and include these and some other references.*

Total electron content variations have been proven a powerful tool in the detection of ionospheric signatures of ULF waves at high latitudes as well as data from ionosondes exploiting the radio-wave reflecting properties of the ionosphere, as it is detailed by Watson et al. (2015). It is not clear to me and perhaps the reader how the results of Watson are different from those of Kozyreva et al. (2019) briefly mentioned in line 29. Nor the difference with those of Pilipenko et al. (2014b) derived from data collected during a different magnetic storm.

*This analysis will be included into the Introduction section and to Discussion. We plan to give a more thorough analysis of observational results in the Introduction section and of physical mechanisms in the Discussion.*

The following publications could be added to improve the placement of this work in the context of existing literature:
- Baddeley et al. (2005), On the coupling between unstable magnetospheric particle populations and resonant high-m ULF wave signatures in the ionosphere, Annales Geophysicae, https://doi.org/10.5194/angeo-23-567-2005
- Buchert et al. (1999), Ionospheric conductivity modulation in ULF pulsations, Journal of Geophysical Research, https://doi.org/10.1029/1998JA900180

*The references will be added to the MS*

In lines 31 and 32, the authors note that the association of waves with moderate amplitudes with variations of the foF2 critical frequency have not been studied. However, how their amplitude is defined as moderate is not described nor later in the manuscript. As mentioned in the title of the manuscript, the reader is waiting for more details on these moderate geomagnetic pulsations, in my mind.

*Thank you very much for this comment. In the next version the data analysis will be improved and a classification of the intervals in accordance with spectral power density at frequencies of PSD local maxima will be added. This will allow to quantify such terms as "moderate".*

In lines 62 and 63, could the authors explain in quantitative terms how high the signal intensity at the reflection boundary should be as well as the amplitude ratio of the signal intensity at the reflection boundary to the power above it? Later, in lines 68 and 71, the authors note that a threshold for the time derivative of the foF2 critical frequency is calculated from the variance over a time interval of length t1. Is the variance of the foF2 critical frequency meant? How is the length of the time interval t1 defined?

The description of the approximation procedure will be extended. Besides, the parameters values used as the initial point of approximation will be added as a supplementary file.

Section 2.2 would benefit from an ionogram on which the described method has been used to detect the ionosphere F2 region critical frequency, clearly illustrating the new method for the foF2 critical frequency automated detection.

Figure 1, its capture, and the text explaining the procedure will be improved to make the detail of the approximation procedure  clearer.

In Figures 4 and 7, it would be worthwhile to note the frequency of the primary and secondary maximum in power and provide further explanation at which frequency the coherence is taken for the statistics provided in Section 3.1.2.

The explanations will be added. Actually, in the present version, there is a difference in the examples, where 2 frequencies are used and statistical results where only the first spectral maxima are analyzed. This point will be improved in the revised version.

In Section 3.1.1, in addition to the details offered for the two intervals in March and July 2015, the two examples could be utilised to introduce the criteria set for selecting similar events for subsequent statistical analysis.

The classification of events will be improved. Really, in the present version, not identical criteria are used at different stages of data analysis. The choice of event class is not random but it may be difficult to discriminate between different types of events taken for comparison with coherent b-foF2 events in each case. The explicit classification will be given in the beginning of the Data processing section.

In Figures 9, 10, 12 and 13, as these are described in Section 3.1.2, what does "occurrence" and the symbol "D" mean in this context? Do the authors refer to "probability of occurrence"?

Yes, that is the empirical probability density, the term will be explicitly explained in the text

As they stand, the conclusions reached and briefly summarised in the first paragraph of Section 4 of this manuscript are a bit vague. Although it is suggested that this study is focused on variations of the ionosphere's critical frequency foF2 during quiet and moderately disturbed geomagnetic conditions, the most favourable values of the Dst index lay between -100 and -50 nT. Under such conditions, how often would it be expected to detect events are associated with ULF geomagnetic pulsations? How would the low occurrence rate (3%) of coherent events change if periods of highly disturbed conditions or quiescence were excluded? Please also consider commenting on the solar wind conditions that are favourable for the occurrence of coherent events and specifically, provide the range of solar wind speed and dynamic pressure values.

The new classification of all the intervals analyzed will give answers to all these questions. You are absolutely right, that in the previous version of our MS, the problems caused by the method of foF2 detection from the ionogram in the disturbed ionosphere can hardly be discriminated with the ionospheric Pc5/Pi3 occurrence probability. In the next version we shall limit ourselves with the disturbance levels, for which the detection procedure is valid and concentrate only on the intervals when quality of foF2 detection

allows for the spectral analysis. For these intervals, we shall analyze the specific features of high coherent b-foF2 pulsations and space weather conditions favorable for their occurrence. Probabilities of coherent b-foF2 pulsations under favorable conditions will be given explicitly in the text.

Lastly, there are inconsistencies in the referencing style and specifically, on page 9 and 10, the year of publication in Mager et al. (2013), Min et al. (2017) and Viall et al. (2009) should be moved to the end of each reference.

The references will be corrected

---

## Author Response (AR1)

We thank both referees for the helpful comments. Point-by-point answers are given in blue.

**Anonymous Referee #1**

Authors: N. Yagova et al.
Title: Even moderate geomagnetic pulsations can cause fluctuations of foF2 frequency of the auroral ionosphere.
The authors investigate the relation between colocated, simultaneous fluctuations in the F2 critical frequency and geomagnetic time series. They developed an automated method for inferring foF2 frequency from the ionograms. Events with foF2 frequencies modulated in the Pc5/Pi3 frequency range are analysed. The properties of a subset of events with coherence greater than 0.5 are compared to the average properties of the whole population of the events. The authors found that coherent events favour moderate geomagnetic conditions and show typical features of field line resonances. At the same time, it is noted that the automated detection of foF2 is not applicable to disturbed conditions. The paper, in general, is difficult to read and follow mainly because of its poor language. We strongly recommend the authors to use some spell-check tool to improve the quality of the presentation of their thoughts.

Thank you very much for your comments. We have rewritten the paper and have tried to improve the language.
A figure illustrating the automated
detection of the critical frequencies would be helpful for the reader.

Figure 1(a) now includes parameters of the approximation and the result of foF2 reconstruction.

 The focus of
the paper is on coherent events, however, neither the 'event' is exactly defined, nor the significance level of the calculated coherence is given.

The significance levels have been added to the paper, and a more detailed classification of the events is now given. This allows to clarify the basis for comparison of the coherent magnetic pulsations with the foF2 pulsations on each step of the data analysis.

There is also some inconsistency
in the paper about at what frequency the coherence is taken (f1 vs f2: statistics
in Fig 10 vs. case studies ). Results presented in Fig 11 again suggest a link to f2 (at least based on the case studies).

Now all the frequencies are included into analysis. An additional criterium of frequency correspondence between foF2 and Bx pulsations is included and presented in Figure 10, and described in the text (lines **212-224**). All frequency information is now presented in Figure 10.

The relative occurrence of coherent events is very
low (_ 3%). The statistics support that coherent events tend to occur under moderately
disturbed geomagnetic and interplanetary conditions. However, the significance
of this result is not clear due to 1. the low relative occurrence of coherent events, 2.
the unknown significance level of coherence, 3. the limitations of foF2 detection under
disturbed conditions, 4. the applied normalisation on which limited information is
given. It was also not investigated how often the coherent events show up provided the
conditions are favourable (moderate disturbance). Without this information the study is
not complete and cannot be judged.

This summary of the problems in data analysis is really very important. Now the data
analysis is redone. We use the following classification of time intervals:
1) All the intervals, during the intervals of observations to analyze space weather
   parameters  (referred as "All" or group1 in the text)
2) all intervals, for which spectra of geomagnetic variations are calculated  (Pc5/Pi3, group
   2 ). In fact, the IMAGE data are available for the absolute majority of intervals. To avoid
   possible influence of seasonal and diurnal variations, we use for comparison the data
   recorded during the same months and UT intervals as foF2 fluctuations (8-14 UT, April
   2014 – end of 2015).
3) The intervals, for which foF2 spectra can be calculated (foF2, group 3)
4) Intervals with the over-threshold coherence level (coherent foF2-Bx, group 4)

The description of data used for statistical analysis is given in the text (lines **136-155)**. The results
are presented in Figures 9-14 and in the text of Section 3.1.2.
We use the combination 1-3-4 for analysis of dependence of foF2 fluctuations and Space Weather
and the combination 2-3-4 for analysis of inter-relation between foF2 and geomagnetic pulsations.

1. As a result, the present version contains a more clear interpretation of the difference in
   pulsation properties and space weather parameters found in the previous version of the
   MS between the coherent foF2-Bx pulsations and the background. Our analysis has
   shown, that it is the result of the two factors. The first one is related to the possibility to
   register foF2 with time resolution enough for spectral estimates and as a whole it
   corresponds to undisturbed conditions. The second one provides the conditions for
   coherent foF2-B fluctuations. They are registered under enhanced auroral activity, and
   specific properties of geomagnetic pulsations. A detailed description is given in the text
   (lines **225 - 291**)

2. The number of analyzed events are given for all the statistical studies (lines 151-152) .
3. Dispersion and confidence levels of coherence are added (lines 159-164, 176-177).

I recommend a major revision. Below I give a list
of my minor comments in two groups. The first group relates to science, the second to
the language. The latter is far from being complete. It would have been a long list.
Minor comments on the scientific content

We have tried to improve the text in the accordance with the minor remarks listed below. Specific remarks are given below to some questions which need more explanations.

l 97: "about 10 nT and 0.08 MHz": Revise these values based on Fig 3!

The average values are corrected, and the maximal amplitudes are added (lines 171-172).

l 100 what is the significance level for the coherence values in these calculations?

The estimates for coherence dispersion and confidence intervals are added (lines 176-177)

l 112: "about 80 nT and 0.08 MHz": although geomagnetic variations are several times greater here then for event 1, the foF2 variations are smaller. Comment?

Our analysis of pulsation properties on the ground has given no clear answer about a factor which controls different amplitude ratios for these two events. Unfortunately, no satellite data at nearly conjugated positions are available for the intervals analyzed. A hypothesis about possible reasons of this difference is now given in the Discussion (lines **325-348**).

l 126: A MLT distribution of occurrence of the foF2 variations –> The MLT distribution of the occurrence of foF2 variations Under what conditions? What criteria define an event?

Now the statistical results are described in a more clear way in accordance with the classification of intervals.

l 128: "frequency distributions of geomagnetic and foF2 pulsations": in general? I guess the distribution is based geomagnetic pulsation events simultaneous with foF2 events.

This description is rewritten

Figure 9: What is D (vertical axis)? Relative occurrence

Now, for all the statistical results empirical probability function P is used.

Figure 10: Why the distribution of the first spectral peaks is presented. In your example events f2 has the higher coherence and corresponding Psw fluctuations. Are not your examples presented typical for the coupling between foF2 and geomagnetic variations?

Now, a new technique with all the spectral maxima taken into account is applied. A new parameter, indicating, whether frequencies of foF2 and Bx spectral maxima are inter-related is introduced. A description of the method and results is given in the text (lines **207-224**)

l 131: "spectral coherence at SOD" : at what frequency? coherence at f1?

l 132: Give the significance level!

Figure 10 b) Mark the significance level in this plot!

This part of the MS is re-organized. We use the other presentation of data. Now the frequency information is given in Figure 10. The dispersion of coherence estimates is discussed in Section 2.

l 142: some information on the derivation of the weight functions and how they applied to normalize the data is needed

The classification of intervals and the way to suppress seasonal and diurnal variation is changed and described in the text (lines **136-155**)

l 153: "the 4-day minimum Dst and 6-hour maximal AE": intervals centred or preceding the coherent event?

All the parameters are given for the preceding intervals. The details are given in the text (lines **126-135**)

l 158-162 Do coherent events occur under severely disturbed conditions, just they cannot be observed? Or they do not occur under those conditions at all? How does this observational limitation affect your conclusions?

A more detailed classification of events has given a more clear discrimination between the limitations caused by a poor foF2 detection under disturbed conditions and the difference between coherent foF2-Bx pulsations and the general population of foF2-Bx cross-spectra. The results of this analysis is presented in the Figures 10-14 and in the text (lines 207-291). As for the ionosonde data, Pc5s in the F layer can be recorded under extremely disturbed conditions only rarely, because of blanketing or absorption below. This leads to the situation when case studies of rare Pc5 events may be possible, but the amount of data is not enough for statistical analysis.

l 165: "3-hour mean values of BZ and V and 3-hour maximal value of _Psw": in which interval? (same issue as above)

Again, the preceding intervals are used

l 181: "Amplitude of SW dynamic pressure fluctuations show an association with occurrence of coherent foF2 _ B pulsations": only 2 examples were presented in favour of this statement. Figure 13 b) does not yield any information on the spectral content of the pressure variations, and hence their relation to the coherent frequency. You seem to focus your statistics on f1 (first peak, e.g. Fig 10 a; coherence at f1), while your 2 examples had their relation with the SW pressure variations at f2.

The data of SW dynamical pressure have many gaps. That is why we only qualitatively consider some example events and use a standard deviation of Psw fluctuations instead of PSD.

l 185: refer to your observations relevant presented in Fig 11. and show how they support the FLR nature of the coherent subset

FLR properties of coherent foF2-Bx pulsations for the event 1 are shown in Figure 15 and described in the text (lines **300-305**).

l 193: "The picture changed dramatically": be more specific!

Thank you very much for the help with the text. We have taken all the comments into account.
Further comments:
l 1: "variations of the critical frequency": maybe "modulation" of the critical frequency could also be used here
l 1: o-mode radiowave –> o-mode radio waves
l 2: in 1–5â ˘ AL'mHz –> in the 1–5â ˘ AL'mHz
l 4: delete "daytime Pc5/Pi3 geomagnetic pulsations and" [foF2 is obviously not detected in geomagnetic pulsations]
l 6: at SOD station –> observed at SOD station
l 6: with the data of a station pair located at the same magnetic meridian –> using the data of a station pair located along the same magnetic meridian
l 8: Meanwhile, –> "At the same time," OR "However,"
l 8: "the analysis of geomagnetic and foF2 variations show intervals with noticeable coherence for both horizontal components" –> "the analysis of geomagnetic ad foF2 variations shows intervals of significant [OR remarkable] coherence with both horizontal geomagnetic components" [foF2 does not have any components]
l 11: averaged –> the average
l 11: coherent to –> coherent with
l 13: show –> shows
l 14: show –> shows [I suggest to use some synonym of 'show', such as 'reveal', 'indicate'. Use an online Thesaurus for finding synonyms]
l 19: Majority of publications are based on the radar observation –> Majority of publications on the topic are based on radar observations
l 20: of electron concentration at certain altitude –> of the electron concentration at a certain altitude
l 23: with mainly compressional mode of MHD wave in the magnetosphere –> with mainly compressional mode magnetospheric waves
l 26: An effect of TEC modulation by ULF wave –> The effect of TEC modulation by a ULF waves
l 27: and zones –> and also from zones
l 28: observed pulsations –> the observed pulsations [a large number of articles are missing from the text, check!]

l 31: the recovery phase of the magnetic storm –> the recovery phase of a magnetic storm

l 33: aimed on variations –> aimed at comparing variations

l 37: It makes an ionogram –> It obtains an ionogram recording

l 40: 10 s sampling rate –> 10 s sampling period/interval

l 40: and we also use the data of the MAS station, which is a part of IMAGE –> we also use data of recorded at MAS station of the IMAGE network

l 42: To analyzed –> To analyze

l 43: and also Dst and AE indexes are used –> as well as Dst and AE indexes

l 46: with quality and time resolution enough –> with good quality and time resolution is enough

l 56: for the reader's sake refer to your Fig 1 here.

l 58: Lorentsian –> Lorentzian

l 60: 235 km –> 235 km.

l 60: Coefficients f1, _f = f2 _ f1, k, and _ are found as a result of fitting procedure, described below. –> A fitting procedure described below is used to find f1, _f = f2 _ f1, k, and _. [f1,_f, _ are not coefficients] What are the meaning of f1 (I guess f at h1) and f2?

l 61: boundary is determined as a line –> boundary consists of a set of (h,f) points

l 62: Signal intensity I at the boundary should be high –> Signal intensity I is high

l 63: Amplitude ratio R of the signal intensity at the boundary line to the power above it should also be high –> The contrast between the peak and the background (characterized by the amplitude ratio R) is high [or similar, your version is confusing. Intensity to power ratio called amplitude ratio... It is not clear what is 'above'. At higher frequency?]

l 64: As four fitting factors are used –> We then fit Eq(1) to the detected boundary points. As four fitting factors are used

l 64: organized and a parameter –> organized. A parameter

l 65: over the "cross" in space of parameters –> over the parameter space [?]

l 65: where x is a point in the space of parameters, and i is a parameter number –> where 'x' is a point in the parameter space, and 'i' identifies the parameter [and what is c? ]

l 64: Give a representative example, e.g. the values of the parameters used to derive the fits presented in Fig 1!

l 68: time dependence f(t): Do you mean the time dependence foF2(t)?

l 69: give a typical value of t1!

l 71: the other [??? or another]

l 73: Examples of approximation curves are given in Figure 1:

Figure 1: Complete the figure caption by including "the fitted curves are plotted over the ionograms in yellow" or similar. Add a reference to the fitting curves in Fig 1 in the main text, as well.

l 76: pictures –> plots

l 79: Note, that the ionograms are rotated by 90â °U ¿e in respect to usual f _ H presentation:

This sentence should come earlier! (with respect to)

l 80: foF2 –> foF2 values

l 84: Statistical analysis: Statistical analysis of what?

l 84: interval –> intervals
l 85: We studied –> We studied the effect/influence of …. on…
l 87: resolution, enough –> resolution high enough
l 90: Cross-spectra are calculated for foF2 variations, on one hand, and components of the geomagnetic field pulsations, on the other hand. –> Cross-spectra are calculated between foF2 variations and components of the geomagnetic field pulsations.
l 100: "at low frequency part of spectrum f < 2 mHz" –> "in the low frequency part (f < 2 mHz) of spectrum"
l 101: peak with maximal $\gamma2 = 0.6$ –> peak with $\gamma2 = 0.6$
l 129: "with frequencies (f1 > 3.7 mHz)" –> "with frequencies above 3.7 mHz
l 130: "The distribution of Pc5/Pi3 intervals over foF2 _ b spectral coherence at SOD are shown in Figure 10b for two" –> "The histogram of the foF2 _ b spectral coherence at SOD is shown in Figure 10b for the two"
l 137: "a question arises about the pulsation properties and external parameters, favorable for their occurrence": rephrase!
l 138: "the geomagnetic pulsations" –> "a subset of the geomagnetic pulsations"
l 139: " with all the intervals, selected" –> " with all the events selected"
l 142: "calculated with the weight functions, which are found from" –> "calculated with weight functions derived from"
l 143: "coherent and pulsations and averaged" –> "coherent pulsations and averaged"
l 151: indexes –> indices
l 159: "limited by" –> "limited to"
l 186: "in coherent foF2 _ Bx pulsations" : delete. This information is already given earlier in the sentence.
l 197: "For the first time, a statistical study of foF2 variations in Pc5/Pi3 range and their relation to geomagnetic pulsation in the conjugated position at SOD station and its spatial distribution along a magnetic meridian." Check the sentence (missing predicate).

**Anonymous Referee #2**

In their manuscript "Even moderate geomagnetic pulsations can cause fluctuations of foF2 frequency of the auroral ionosphere", Yagova et al. explore variations of the ionosphere F2 region critical frequency (foF2) and ultra-low frequency (ULF) waves in the Pc5 and Pi3 frequency bands detected at auroral latitudes.
Using ground magnetometer and ionosonde data spanning years 2014 and 2015, the authors examine the power, coherence and phase difference of perturbations in the daytime ionosphere and Pc5/Pi3 geomagnetic pulsations, distinguishing a subset of events during periods of magnetic quiescence and moderate magnetic storms with coherence greater than 0.5 from ULF wave signatures in the ionosphere observed under conditions favourable to strong geomagnetic storms. This extends previous studies by Pilipenko et al. (2014a and 2014b) that considered ULF wave-driven oscillations in the ionosphere F2 region during strong and small magnetic storms.
Furthermore, the manuscript presents a new methodology to the automated detection of the foF2 critical frequency from ionograms that could be of interest for the research

community working on determining factors that influence the amplitude and phase of perturbations in the ionosphere as these are detected on the ground. There are, however, several issues that hinder my recommendation of this manuscript for publication in Annales Geophysicae in its present form.

There are major issues with the English language use, several typographical errors and in general, it is poorly written making it difficult to understand the scientific rationale behind this study.

Thank you very much for the comments. We have re-examined the MS and have tried to improve the language.

For example, in line 19, it reads: "Modulation of ionospheric parameters by Pc5 pulsations was reported : : :", without detailing which parameters are meant here. In the same line, it goes on to say: "Majority of publications are based on the radar observation : : :" (which would more correctly read "The majority of publications are based on radar observations : : :"), without making it clear to which publications the authors refer.

The Introduction is rewritten and we tried to make the description  of publications  more specific

It would be worthwhile to establish in the Introduction the need for a study such as the present by listing past publications focused on perturbations in the ionosphere driven by ULF waves. Early results on geomagnetic pulsations in the ULF wave frequency range associated with total electron content (TEC) fluctuations date back to 1976 and include the following:
- Davies & Hartmann (1976), Short-period fluctuations in total columnar electron content, Journal of Geophysical Research, https://doi.org/10.1029/JA081i019p03431
- Okuzawa & Davies (1981), Pulsations in total columnar electron content, Journal of Geophysical Research, https://doi.org/10.1029/JA086iA03p01355

In the previous version, we have briefly mentioned only auroral Pc5 pulsations, while the papers by Davies (1976) and Okuzawa (1981) were devoted to Pc3-4 pulsations at lower latitudes. In the revision we have  extended the Introduction section.

Total electron content variations have been proven a powerful tool in the detection of ionospheric signatures of ULF waves at high latitudes as well as data from ionosondes exploiting the radio-wave reflecting properties of the ionosphere, as it is detailed by Watson et al. (2015). It is not clear to me and perhaps the reader how the results of Watson are different from those of Kozyreva et al. (2019) briefly mentioned in line 29. Nor the difference with those of Pilipenko et al. (2014b) derived from data collected during a different magnetic storm.

This analysis is included into the Introduction section and to Discussion. The difference of physical mechanisms providing different TEC to geomagnetic amplitude ratios is now discussed in the text (lines 325-348).

The following publications could be added to improve the placement of this work in the context of existing literature:
- Baddeley et al. (2005), On the coupling between unstable magnetospheric particle populations and resonant high-m ULF wave signatures in the ionosphere, Annales Geophysicae, https://doi.org/10.5194/angeo-23-567-2005
- Buchert et al. (1999), Ionospheric conductivity modulation in ULF pulsations, Journal of Geophysical Research, https://doi.org/10.1029/1998JA900180

The references have been added to the MS

In lines 31 and 32, the authors note that the association of waves with moderate amplitudes with variations of the foF2 critical frequency have not been studied. However, how their amplitude is defined as moderate is not described nor later in the manuscript. As mentioned in the title of the manuscript, the reader is waiting for more details on these moderate geomagnetic pulsations, in my mind.

In the presented version the data analysis is redone. Now the dependence of foF2-Bx coherence on PSD of geomagnetic pulsations is discussed in the text (lines 251-256 and Figure 12) .

In lines 62 and 63, could the authors explain in quantitative terms how high the signal intensity at the reflection boundary should be as well as the amplitude ratio of the signal intensity at the reflection boundary to the power above it?
Later, in lines 68 and 71, the authors note that a threshold for the time derivative of the foF2 critical frequency is calculated from the variance over a time interval of length t1. Is the variance of the foF2 critical frequency meant? How is the length of the time interval t1 defined?

The description of the approximation procedure is extended and the parameters used are mentioned in the text (lines 86-98).

Section 2.2 would benefit from an ionogram on which the described method has been used to detect the ionosphere F2 region critical frequency, clearly illustrating the new method for the foF2 critical frequency automated detection.

Figure 1, its capture, and the text explaining the procedure is improved to make the detail of the approximation procedure clearer.

In Figures 4 and 7, it would be worthwhile to note the frequency of the primary and secondary maximum in power and provide further explanation at which frequency the coherence is taken for the statistics provided in Section 3.1.2.

Now the statistical analysis is extended and all frequencies of coherence maxima are included into statistics

In Section 3.1.1, in addition to the details offered for the two intervals in March and July 2015, the two examples could be utilised to introduce the criteria set for selecting similar events for subsequent statistical analysis.

The classification of events is improved. The explicit classification is given in the Data processing section (lines 136-155).

In Figures 9, 10, 12 and 13, as these are described in Section 3.1.2, what does "occurrence" and the symbol "D" mean in this context? Do the authors refer to "probability of occurrence"?

Now, for all the statistical distribution empirical probability function P is used.

As they stand, the conclusions reached and briefly summarised in the first paragraph of Section 4 of this manuscript are a bit vague. Although it is suggested that this study is focused on variations of the ionosphere's critical frequency foF2 during quiet and moderately disturbed geomagnetic conditions, the most favourable values of the Dst index lay between -100 and -50 nT. Under such conditions, how often would it be expected to detect events are associated with ULF geomagnetic pulsations? How would the low occurrence rate (3%) of coherent events change if periods of highly disturbed conditions or quiescence were excluded? Please also consider commenting on the solar wind conditions that are favourable for the occurrence of coherent events and specifically, provide the range of solar wind speed and dynamic pressure values.

The new classification of all the intervals analyzed is now used. You are absolutely right, that in the previous version of our MS, the problems caused by the method of foF2 detection from the ionogram in the disturbed ionosphere can hardly be discriminated with the ionospheric Pc5/Pi3 occurrence probability. In the present version we show the difference in average space weather parameters for all the 21 months analyzed and for the foF2 intervals, i.e. those with foF2 quality and time resolution enough for spectral estimates in Pc5 range.  For these intervals, the specific features of high coherent b-foF2 pulsations and space weather conditions favorable for their occurrence are analyzed. The results are presented in the Figures 11-14 and in the text (section 3.1.2). Probabilities of coherent b-foF2 pulsations under favorable conditions are given explicitly in the Figures and discussed in the text (lines 256-292).

Lastly, there are inconsistencies in the referencing style and specifically, on page 9 and 10, the year of publication in Mager et al. (2013), Min et al. (2017) and Viall et al. (2009) should be moved to the end of each reference.

The references are corrected

[revised manuscript text omitted]

---

## Author Response (AR2)

Dear Editor and referees, we have revised the manuscript according to referees' s remarks. Their reports and our point-by-point answers marked by color are given below.

**Referee 1**
**Suggestions for revision or reasons for rejection (will be published if the paper is accepted for final publication)**
The paper has improved a lot through the latest revision. My comments have been addressed adequately. The science content is now much more sound than in the previous version. The relations found between the occurrence of coherent fluctuations in geomagnetic field and foF2 with the parameters investigated are rather weak. According to 10b, the improvement in the frequency correspondence (absolute difference) between magnetic and foF2 pulsations for the coherent events is less than 5% compared to the same difference calculated for all events. This needs some explanation and discussion.

Actually, the result in Figure 10 compares frequency distributions not between coherent intervals (group 4) and non-coherent ones, but between all the intervals with geomagnetic pulsations (group 2) and those, when foF2 fluctuations were also registered (group 3). Figure 10 (b) presents the results of comparison of mean square difference between frequencies of spectral maxima $\Delta_{f2}$ within group 3 for the spectra of simultaneous geomagnetic and foF2 pulsations and those calculated at random (generally, not equal dates). The explanation of this point is extended (lines 220-234)

The relative occurrences of the coherent events (vs different parameters) are presented at two levels of coherence. For me, it seems unnecessary and rather confusing then helpful. I suggest to keep only the distributions at the gamma_squared = 0.5 level. At higher level, the occurrence rate of coherent events remains mostly below 5%. It just means that highly coherent events are rare.

The second threshold for coherence is excluded

However, the main problem with the current version of the paper is that it is still very difficult to read.
Firstly, because the organisation of the paper is not reader-friendly. Section 2.3 (Pre-processing), especially from line 143 is a mere listing of the spectral, statistical quantities applied later in the paper. Without giving examples, it is difficult to follow what is the function and purpose of the different quantities in

the data analysis, and also the definition of the quantities is sometimes unclear. (e.g. "the fraction of intervals Rγ with over-threshold maximal γ2 is presented").

The information given in the previous version in Section 2.3 (lines 148-155) is now presented in Section 3.

Secondly, because of the confusing use of the terminology and notation. Different terms are used for the same thing, however, it is not explained to the readers that they refer to the same thing. E.g. "a fraction" (that is itself very confusing) is also called "coherent ratio" in the meaning of 'relative occurrence of coherent events'. Another example: "meridional distribution" vs. "south-to-north ratio" in the meaning of 'frequency dependence of the south-to-north ratio' or 'south-to-north spectral ratio.

In the revised version, the only term "Meridional PSD ratio" is used

It is also confusing, that the same letters are used to denote very different things. E.g. P means probability, PSD, and also PSD peak (although in the latter- $P_{f,b}$ - meaning it is always referred to as PSD maximum in the article. 'PSDmax' or similar would be more natural and understandable choice for the latter).

Some indices seems unnecessary, e.g. since PSD (except for the very last figure) is only calculated from magnetic signals, so $P_{f,b}$ or $PSD_{f,b}$ could be simply PSDmax.

Now all the PSDs are given in the Figures explicitly and some of indices are excluded

Both 'F' and 'f' are used for frequency.

The explanation for different notations is included in the beginning of section 2 (lines 89-91).

Phi denotes both phase and magnetic latitude.

Actually, different notations are used for phase, $\phi$ and magnetic latitude $\Phi$. However, they really look similarly in Figures 11 and 15 and we have used the explicit description $PSD_{SOD}/PSD_{MAS}$ in the axis labels.

'R' stands for PSD ratio and relative occurrence, etc.

In the revised version, we use R for the PSD ratio, while "Rel. Oc." is used for relative occurrence in axis labels of Figures 9 and 12-14.

Finally, the English of the paper is still very poor. I add a list of some of the typos I found. This list is very far from being complete!

27 filed > field +
73 …AE > … AE indices. +
75 :: > : +
78 at, in > between, at +
106 foF2 which > foF2 for which the +
136 for the > for +
152 at > of +
161 the its > its +
201 the distributions of the parameters of foF2 variations with the magnetic local time > MLT-dependence of the occurrence rate +
202 occurrence of foF2 interval [confusing phrase] removed
204 a fraction of > the occurrence rate of coherent events [?] The phrase is rewritten
205 bb maximum: what is bb? Two threshold values is excluded from the text and Figures, and this phrase is omitted
205, 206: the noon > noon +
Fig 11 in the legend 1 should be 2 or the text should be adjusted - done
242 which it is [?]. The paragraph is rewritten.
253: 3 times higher: I cannot see what you mean – the $PSD_{max}$ values for both groups are included into the text
261: A difference … is seen: I cannot see it
261 probablity of the … interval [?] The phrase is rewritten
Fig 13 R_gamma : the Greek letter should be used here +

**Referee2**

**Suggestions for revision or reasons for rejection (will be published if the paper is accepted for final publication)**
This revised manuscript explores the statistical behaviour of ionospheric variations in the frequency range of 1 -5 mHz and simultaneous low-frequency geomagnetic pulsations observed at auroral latitudes during periods of quiescence but also under moderately disturbed geomagnetic conditions. Using a newly developed method for the automated detection of ionospheric F2 region's critical frequency, foF2, from ionograms, Yagova et al. investigated time intervals where fluctuations of the ionosphere's FoF2 frequency are accompanied by similar temporal variations of the magnetic field's northward component as this has been measured by ground magnetometers. They identified several cases of high coherence indicating a convincing relationship between low-frequency geomagnetic pulsations and the modulation of the ionosphere's F2 layer. They also looked into favourable solar wind conditions to find that solar wind pressure pulses, known to

excite magnetospheric ULF waves, are among the possible drivers of both geomagnetic pulsations and particle modulation.

The presentation of their methodology and analysis' results have greatly helped the support of the manuscript as well as its readability. This revised manuscript has a few issues that I would like to see addressed to the editor's satisfaction before I could recommend publication of this manuscript.

In line 1 of the Abstract, it would be more convenient for the reader to know that the coordinates provided for the Sodankyla Geophysical Observatory are the geographical coordinates rather that to have to look this up in Table 1. done

In line 3 of the Abstract and throughout the manuscript, the authors talk about Pc5/Pi3 waves. I would caution the authors against referring to Pc5 and Pi3 waves together, when Pc5 waves are considered to have a sinusoidal wave form and last longer than Pi3 waves that have an irregular wave form. Pi3 waves can be the ground counterpart of Pc5 waves with moderately high-m numbers observed by spacecraft, as it was found by Vaivads et al (2001), but this study is centred around geomagnetic pulsations observed on the ground. The authors cite several published results of research carried out with focus on either Pi3 or Pc5 waves. Among others, Kleimenova et al. (2002) showed that auroral substorms influence Pi3 activity but Baker et al. (2003) isolated Pc5 waves whose amplitude did not grow nor was damped as is the case of irregular pulsations for their study of Pc5 waves. Furthermore, Posch et al. (2003) adopted a ULF index that allows to minimise the power contribution of large, irregular pulsations along with the total ULF wave power which may include the substorm counterpart of ULF waves. Although, Pc5 and Pi3 waves' frequency falls in the same range, adequate justification for the choice to consider the two types of waves together needs to be provided.

The explanation is now given in the end of the beginning of section 2.3 (lines 127-132). As we do not analyze pulsation types and analyze all the pulsations at frequencies 1-5 mHz, we have change Pc5/Pi3 to Pc5-6/Pi3 throughout the text.

In line 77 of Section 2.1, the authors note that geomagnetic pulsations are searched for in the BX component of the magnetic field measured by ground magnetometers (it should read "Bx"). Later in line 119 of Section 2.3, the symbol bx is used for variations of the same component. However, it is not made clear how these variations have been identified. Has a mean value been calculated that corresponds to the Earth's main magnetic field, and variations been defined as deviations from this mean? Has a different approach been adopted for the derivation of magnetic field variations?

An explanation of different notations for variations of foF2 and B and their absolute values is moved to the beginning of Section 3.1, where filtered time

series are shown in Figure 3 (lines 177-179)

In line 79 of Section 2.1, the authors refer to Table 2 that includes a list of more than 80 intervals that have been selected for further analysis. (It is redundant to provide the same table in the Supplement.)

The double tables are excluded

In Section 2.3, these intervals where foF2 has been retrieved utilising the new methodology have been divided into four groups. Please add group 5 to the list which has not been introduced until in Section 3.1.2

done

and point out the assumptions that have been used in statistical analysis of data covering time intervals of varying length unambiguously.

Actually, the time intervals are of equal length (64 points), and all the analysis is based on the spectra for these standard intervals. The explanation of this point is extended (lines 156-159)

It is also strongly recommended to provide sufficient details of the grouping of foF2 frequency data in 2764 overlapping intervals, as this is mentioned in line 151, to prove the validity of the statistics.

In the revised version of the MS, a more detailed description of intervals with the account taken of overlapping is given in the text (lines 160-170)

For instance, it is not so clear how it was concluded that foF2 frequency fluctuations and geomagnetic pulsations are more likely to occur coherently in the recovery phase of moderate geomagnetic storms.

We use a 4-day minimum of Dst index, i.e. pulsations at the recovery phase are included into analysis. Now an explanation is given (lines 265-273) and the details of time delay distribution at 3 levels of minimal Dst are available in a supplementary file.

In the last two paragraphs of Section 2.3, the authors provide details of the power spectral density calculations and cross-spectral analysis. It is, however, not clear whether the Blackman-Tukey correlogram/periodogram analysis has also been used for the calculation of PSD in the Bx and By component.

This part of Section 2.3 is modified, and the description of spectral estimates is given in more details (lines 152-170)

In line 171 of Section 3.1.1, it would more correctly read "Peak amplitudes of geomagnetic pulsations and foF2 variations are about 8 nT and 0.05 MHz." and in line 183, "Their peak-to-peak amplitude was about 0.7 nPa …". In line 178, it should read "…start time of the interval (12:20 UT)."

Done

On page 16, please consider adding a colour bar in Figure 1 with the three ionograms from the Sodankylä Geophysical Observatory provided as examples. This would also help make clearer the first of the two criteria set for the automated detection of foF2 and specifically, the condition that the intensity of reflection should fulfil.

Done

Please consider providing the units of measurement in line 97 of Section 2.2 for the lower value of the intensity.

The reflection intensity in Figure 1 is given in dB, while for the approximation in Eq.1. linear scale is used (voltage in arbitrary units). The units are now given in the text (lines 95-96) and in the Figure 1 capture.

Please note that there are numerous typos making it necessary to carefully revise the manuscript in terms of spelling and punctuation as well as language. For example,

in line 6 of the Abstract, the authors must mean "automated retrieval of foF2, the critical frequency of the F2 layer from ionograms."  done

In line 27, "filed line resonances" should be "field line resonances".  done

In line 37, the comma after "observed oscillations is, …" should be deleted. +

In line 56, it is not clear to me and perhaps the reader what the term "principle opportunity" means.

This fragment of the text is extended and re-formulated and a direct reference to soft electrons modulation by long-period ULF waves is included (Ren et al., 2019, lines 55-57).

In line 66, "ionosond" should be changed to "ionosonde". +

In line 83, it would more correctly read "variability of the intensity of reflected signals, background noise, sporadic layers and irregularities, broadcast interference, etc."  done

In line 88, it would more correctly read "near linear growth" and, in  done

line 89, "Lorentzian function" instead of "Lorentzian shape function" which makes one think of spectral line shape.    done

In line 164, it should read "$\gamma b2 = 0.64$, $\gamma 2 > 0.375$". This phrase is deleted in accordance with the suggestion of Referee 1 to give only one threshold value for coherence.

Lastly, the format of citations should be revised. For example, in line 25, "(Wright et al., 1997) should be changed to "Wright et al. (1997)" and in line 47, "(Pilipenko et al. 2014b)" should be "Pilipenko et al. (2014b)".  done

References

Saito (1978), Long-period irregular magnetic pulsation, Pi3, Space Science Reviews, doi: 10.1007/BF00173068

Vaivads et al. (2001), Correlation studies of compressional Pc5 pulsations in space and Ps6 pulsations on the ground, Journal of Geophysical Research, doi: 10.1029/2001JA900042

---

## Author Response (AR3)

Dear Dr. Balasis,

We present the revised version of the MS angeo-2020-16 "Even moderate geomagnetic pulsations can cause fluctuations of foF2 frequency of the auroral ionosphere". All the referee's remarks are taken into account. The point-by-point answers are given below in blue. We have tried to improve readability of the manuscript throughout the text. All the changes are indicated in the marked version of the MS. The corrected phrases are highlighted. Besides, the fragments of section 2.3 are underlined to show that it was re-structured.

There are still several issues, including the following:

In lines 2 and 3, the frequency range of geomagnetic pulsations examined in this study is first mentioned. Specifically, it should read "…with frequencies close to those of geomagnetic pulsations in the Pc5-6/Pi3 frequency band (1 – 5 mHz)." Later, in line 6, it should read "... As a rule, the Pc5-6/Pi3 frequency band …". The same notation should be maintained throughout the manuscript.

We have checked the notation throughout the MS and we have changed it in the discussion of statistics. However, at the points where FLR effects for Pc5s were discussed, the pulsations' frequencies definitely show that they are Pc5 waves and the notation Pc5 is used. In citations we followed the authors.

In line 32, it would more correctly read " … devoted to Pc5 wave signatures in total …".

done

In lines 55 to 57, it should read "…six years of Van Allen Probes measurements confirmed the influence of ULF waves in Pc4 and Pc5 frequency ranges on electrons with energies up to the order of 10^2 eV with maximum occurrence around L from 5.5 to 7 (Ren et al., 2019)."

done

In lines 133 and 134, what is written in the previous lines 131 and 132 is essentially repeated.

The second phrase is excluded

In lines 135 to 137, the two indices, AE and Dst, as well as parameters of the solar wind considered in the present analysis are repeated. They were first introduced in Section 2.1.

The phrase about indices is excluded, and the one about IMF/SW parameters is re-formulated and moved to section 2.1.

In line 138, it should read " …analysis, all intervals were classified …".

corrected

In line 141, it should read either "…spectra of geomagnetic pulsations …" or "…spectra of geomagnetic field variations …". The terms "geomagnetic field variations" and geomagnetic pulsations should be adopted in the place of "geomagnetic variations" which it is not clear to me - and perhaps the reader - to what it refers. However, it is often used in this version of the manuscript.

corrected

In line 142, should it read "…intervals, for which spectra of foF2 variations can be calculated (ΔfoF2 intervals …"?

And later, in line 144, should it read "…coherent ΔfoF2-bx intervals, …" even though the explanation of ΔfoF2 and bx are not provided earlier that the next Section 3.1? It is the coherence between

filtered time series of the foF2 frequency and the meridional component of the geomagnetic field B that has been calculated, if I have not confused the notations.

The notation is corrected

Furthermore, the organisation of Section 2.3 could be improved if the first introductory paragraphs (in the current version of the manuscript, it is not clear whether there are two, three or more paragraphs from the beginning until line 137) were followed by details of the data analysis carried out which are presented in lines 152 to 170. The classification of intervals examined that is currently found between lines 138 and 145 should be placed before the statistics for these intervals presented in lines 160 to 162.

The subsection is re-organized, the paragraph structure is revised.

In line 174, it would more correctly read "foF2 frequency and geomagnetic field variations"

done

and later, in line 177, "A high pass filter with cut-off frequency at 0.8 mHzThe time series in Figures 3 and 6 are filtered …".

done

In line 263, it could read "Statistics on ΔfoF2 and bx fluctuations related to different levels of geomagnetic activity are presented in Figure 13. Left panels (13a, c) present the occurrence distribution for all, ΔfoF2 and coherent ΔfoF2-bx intervals (groups 1, 3 and 4)…".

done

In Figure 13, would it be possible to add labels to tick marks other than 100 and 1000 which are currently visible in plots 13c and 13d so that it is easier to follow the description of the statistics in Section ?

done

This list of issues identified is far from inclusive but hopefully indicative of the manuscript's poor readability.

We have tried to improve readability of the manuscript throughout the text. All the changes are indicated in the marked version of the MS.

---

## Author Response (AR4)

Dear Dr. Balasis,

We present the final version of the MS angeo-2020-16 "Even moderate geomagnetic pulsations can cause fluctuations of foF2 frequency of the auroral ionosphere".  Thank you very much for helpful remarks and collaboration at all stages of MS preparation.

With best regards
Nadezda Yagova